# A non-avian dinosaur with a streamlined body exhibits potential adaptations for swimming

Sungjin Lee[1], Yuong-Nam Lee [1✉], Philip J. Currie [2], Robin Sissons [2], Jin-Young Park[1], Su-Hwan Kim[1], Rinchen Barsbold[3] & Khishigjav Tsogtbaatar [3]

Streamlining a body is a major adaptation for aquatic animals to move efficiently in the water. Whereas diving birds are well known to have streamlined bodies, such body shapes have not been documented in non-avian dinosaurs. It is primarily because most known non-avian theropods are terrestrial, barring a few exceptions. However, clear evidence of streamlined bodies is absent even in the purported semiaquatic groups. Here we report a new theropod, *Natovenator polydontus* gen. et sp. nov., from the Upper Cretaceous of Mongolia. The new specimen includes a well-preserved skeleton with several articulated dorsal ribs that are posterolaterally oriented to streamline the body as in diving birds. Additionally, the widely arched proximal rib shafts reflect a dorsoventrally compressed ribcage like aquatic reptiles. Its body shape suggests that *Natovenator* was a potentially capable swimming predator, and the streamlined body evolved independently in separate lineages of theropod dinosaurs.

[1] School of Earth and Environmental Sciences, Seoul National University, Seoul, Korea. [2] Department of Biological Sciences, University of Alberta, Edmonton, AB, Canada. [3] Institute of Paleontology, Mongolian Academy of Sciences, Ulaanbaatar, Mongolia. ✉email: ynlee@snu.ac.kr

The Gobi Desert of Mongolia is the source of non-avian theropod dinosaurs that provided important evidence for brooding behaviour[1], the presence of pygostyles[2], and a long-armed giant omnivore[3]. A recent study of the dromaeosaurid theropod *Halszkaraptor* from the Djadochta Formation of this region revealed its semiaquatic ecology, which is unique among non-avian maniraptorans[4]. Its morphological specializations include a snout with a complex neurovascular network, retracted nares, a dental arrangement for capturing evasive prey, an unusually long neck similar to that of known aquatic reptiles, and horizontal zygapophyses in the cervical and caudal vertebrae[4]. Moreover, the flattened forelimb bones and the proportions of the manual digits of *Halszkaraptor* were similar to birds that use their forelimbs for swimming[4]. However, even though several aquatic adaptations were identified in *Halszkaraptor*, its body shape could not be inferred from the preserved specimen. *Hulsanpes*[5] and *Mahakala*[6], the closest relatives of *Halszkaraptor*, are too poorly preserved or are missing key regions to provide clues about the ecology of this clade.

A new theropod dinosaur, *Natovenator polydontus* gen. et sp. nov., is described based on a well-articulated specimen (Figs. 1, 2, 3a–h, 4a, Supplementary Note 1, and Supplementary Figs. 1–4) from the Baruungoyot Formation at Hermiin Tsav in the southern Mongolian Gobi Desert. This new taxon exhibits anatomical characteristics very similar to the aquatic adaptations in *Halszkaraptor*[4,7]. More importantly, the configuration of its articulated dorsal ribs indicates that it had a dorsoventrally flattened and streamlined body. Because streamlining of the body provides hydrodynamic advantages during swimming[8–12], this particular dorsal rib morphology strongly indicates that *Natovenator* was a capable swimmer, providing the first compelling evidence of a streamlined body in a non-avian theropod dinosaur.

It thus exemplifies the presence of diverse body forms among non-avian theropods. In addition, *Natovenator* helps us understand the body plans of halszkaraptorines because it shares many specialized features with *Halszkaraptor*.

## Results

Dinosauria Owen, 1842
 Theropoda Marsh, 1881
 Dromaeosauridae Matthew and Brown, 1922
 Halszkaraptorinae Cau et al., 2017

**Revised diagnosis**. Small dromaeosaurids that possess dorsoventrally flattened premaxillae, premaxillary bodies perforated by many neurovascular foramina, enlarged and closely packed premaxillary teeth that utilized delayed replacement patterns, reduced anterior maxillary teeth, dorsolateral placement of retracted external nares, greatly elongated cervical vertebrae, anterior cervical vertebrae with round lobes formed by the postzygapophyses, horizontal zygapophyses, and pronounced zygapophyseal laminae in the anterior caudal vertebrae, mediolaterally compressed ulnae with sharp posterior margins, second and third metacarpals with similar thicknesses, shelf-like supratrochanteric processes on the ilia, elongated fossae that border posterolateral ridges on the posterodistal surfaces of the femoral shafts, and third metatarsals in which the proximal halves are unconstricted and anteriorly convex.

*Natovenator polydontus* gen. et sp. nov.

**Holotype**. MPC-D 102/114 (Institute of Paleontology, Mongolian Academy of Sciences, Ulaanbaatar, Mongolia) is a mostly

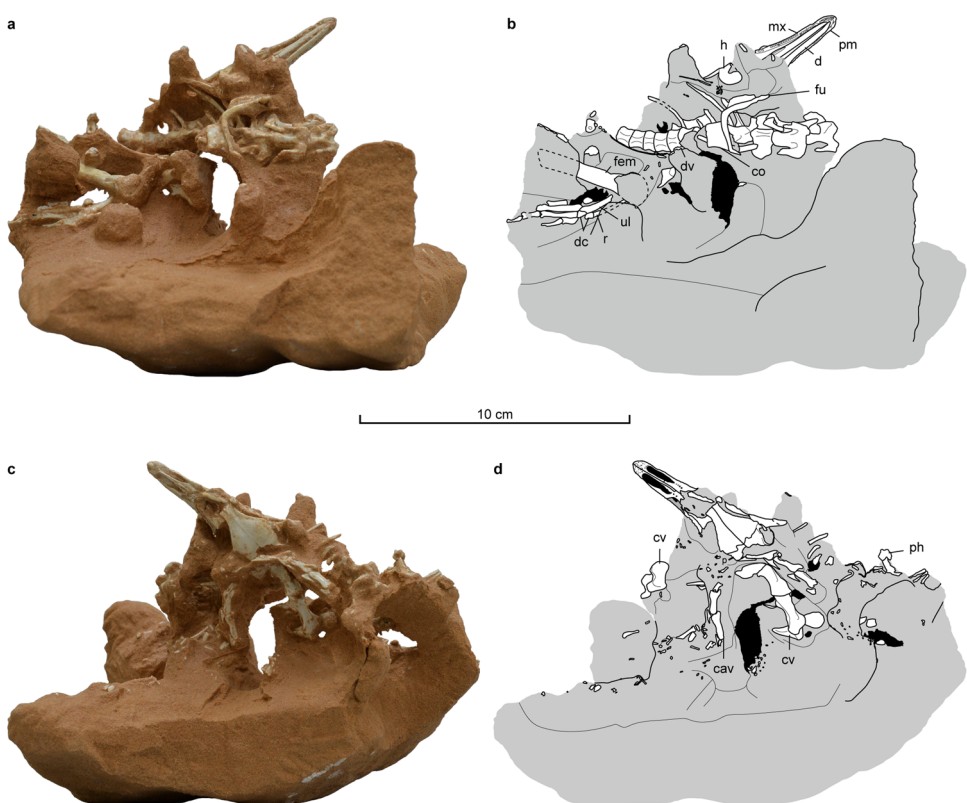

**Fig. 1 *Natovenator polydontus* (MPC-D 102/114, holotype).** Photographs (**a**, **c**) and line drawings (**b**, **d**) of the main block containing most of the specimen in opposite views. cav caudal vertebra, co coracoid, cv cervical vertebra, d dentary, dc distal carpal, dv dorsal vertebra, fem femur, fu furcula, h humerus, mx maxilla, ph phalanx, pm premaxilla, r radius, ul ulna.

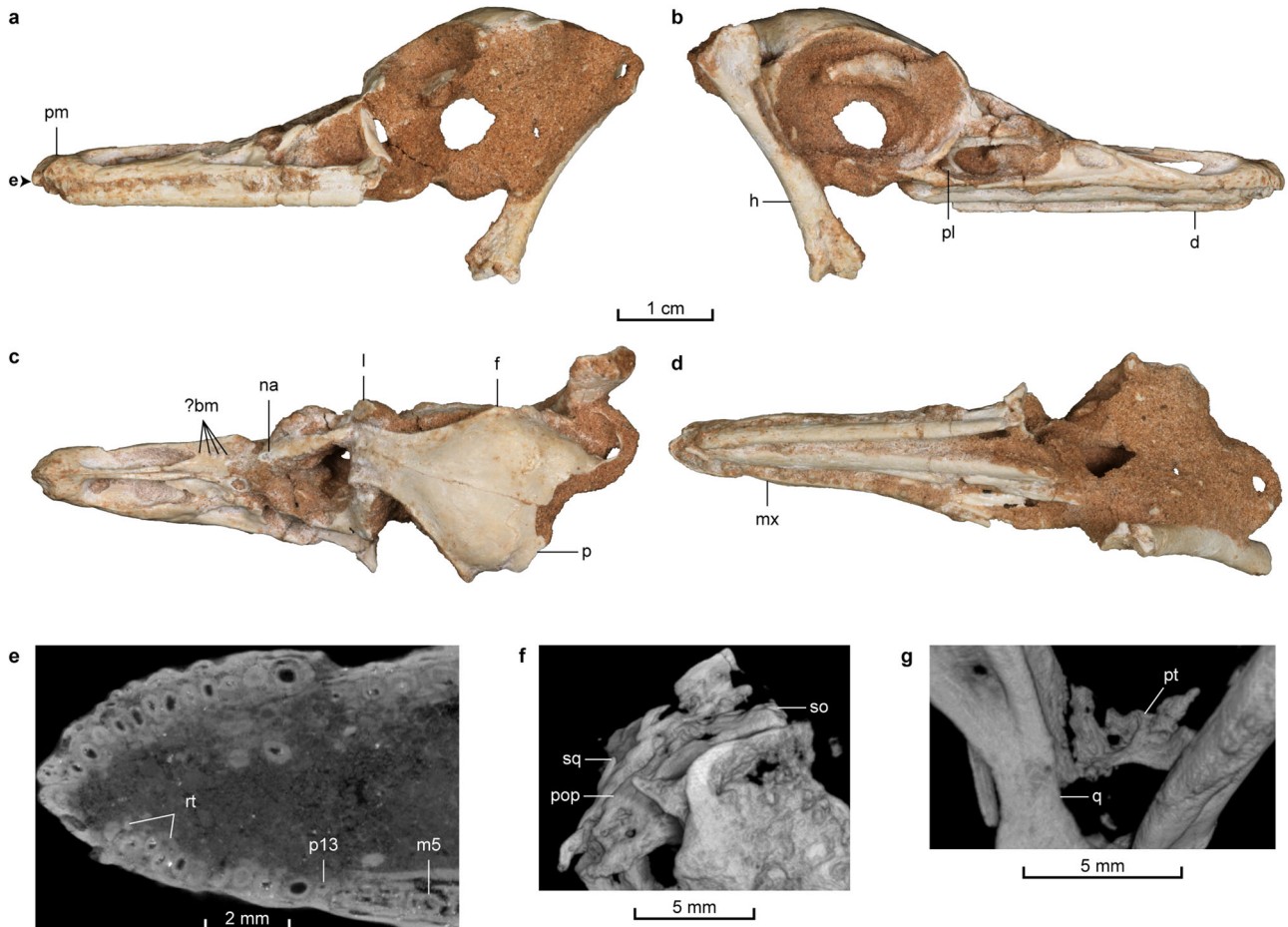

**Fig. 2 Skull of *Natovenator polydontus* (MPC-D 102/114, holotype). a–d** Skull in left lateral (**a**), right lateral (**b**), dorsal (**c**), and ventral (**d**) views. **e** μCT-rendered image sliced at the point marked on **a**, showing a cross-section of the premaxillary and anterior maxillary teeth in dorsal view. **f** Micro-computed tomography (μCT) rendered image of the occipital region in posterior view. **g** μCT-rendered image of the pterygoid and quadrate. ?bm possible bite mark, d dentary, f frontal, h humerus, l lacrimal, m5 5th maxillary tooth, mx maxilla, na nasal p parietal, p13 13th premaxillary tooth, pl palatine, pm premaxilla, pop paroccipital process, pt pterygoid, q quadrate, rt replacement tooth, sq squamosal, so supraoccipital.

articulated skeleton with a nearly complete skull (See Supplementary Table 1 for measurements).

**Locality and horizon**. Baruungoyot Formation (Upper Cretaceous), Hermiin Tsav, Omnogovi Province, Mongolia[13] (Supplementary Fig. 5).

**Etymology**. *Natovenator*, from the Latin *nato* (swim) and *venator* (hunter), in reference to the hypothesized swimming behaviour and piscivorous diet of the new taxon; *polydontus*, from the Greek *polys* (many) and *odous* (tooth) in reference to the unusually many teeth.

**Diagnosis**. A small halszkaraptorine dromaeosaurid with the following autapomorphies: wide groove delimited by a pair of ridges on the anterodorsal surface of the premaxilla, premaxilla with an elongated internarial process that overlies nasal and extends posterior to the external naris, 13 premaxillary teeth with large and incisiviform crowns, first three anteriormost maxillary teeth are greatly reduced and are clustered together with the following tooth without any separations by interdental septa, anteroposteriorly long external naris (about 30% of the preorbital skull length), paroccipital process with a anteroposteriorly broad dorsal surface, elongate maxillary process of the palatine that

extends anteriorly beyond the middle of the antorbital fenestra, pterygoid with a deep fossa on the medial surface of the quadrate ramus, distinct posterolaterally oriented projection on the lateral surface of atlas, absence of pleurocoels in cervical vertebrae (not confirmed in the missing fifth cervical centrum), posterolaterally oriented and nearly horizontal proximal shafts in the dorsal ribs, hourglass-shaped metacarpal II with distinctly concave medial and lateral surfaces.

*Description*. The skull of *Natovenator* is nearly complete, although the preorbital region has been affected by compression and is slightly offset from the rest of the skull (Figs. 1c, d, 2a–d and Supplementary Figs. 1, 2). Near the tip of the snout, the premaxilla is marked by a broad groove. The body of the premaxilla is also dorsoventrally low and is perforated by numerous foramina that lead into a complex network of neurovascular chambers (Supplementary Fig. 1b) as in *Halszkaraptor*[4]. Similarly, the external naris is positioned posteriorly and is level with the premaxilla-maxilla contact (Fig. 2a, b), although it is marginally behind this position in *Halszkaraptor*[4]. It is also dorsally placed compared to those of other non-avian theropods and faces dorsolaterally. The exceptionally long external naris and accordingly elongated internarial process of *Natovenator* (Fig. 2c) are unique among dromaeosaurids but comparable to those in aquatic toothed birds[14] as well as in therizinosaurs[15,16]. The

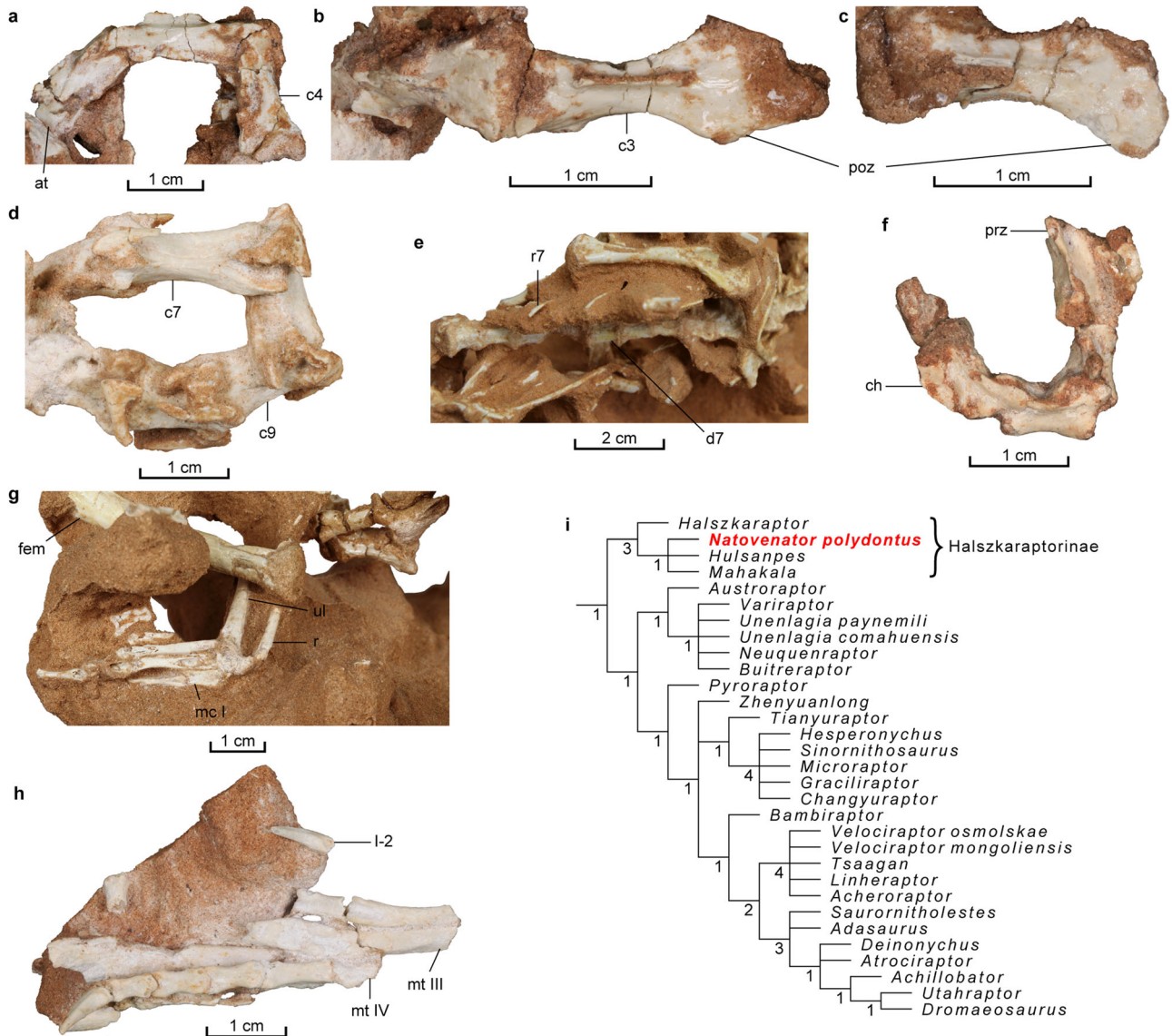

**Fig. 3 Postcranial elements and phylogenetic position of *Natovenator polydontus* (MPC-D 102/114, holotype). a** Anterior cervical vertebrae in left lateral view. **b** Axis and third cervical vertebra in dorsal view. **c** Fourth cervical vertebra in dorsal view. **d** Posterior cervical vertebrae in right lateral view. **e** Dorsal series in right lateral view. **f** Anterior caudal vertebrae in right lateral view. **g** Left forearm elements in medial view and manus in ventral view. **h** Right foot in ventral view. **i** Phylogenetic position of *Natovenator* in Dromaeosauridae. Numbers at each node indicate Bremer support values. at atlas, c3 third cervical vertebra, c4 fourth cervical vertebra, c7 seventh cervical vertebra, c9 ninth cervical vertebra, ch chevron, d7 seventh dorsal vertebra, fem femur, mc I metacarpal l, mt III metatarsal III, mt IV metatarsal IV, poz postzygapophysis, prz prezygapophysis, r radius, r7 seventh dorsal rib, ul ulna, I-2 pedal phalanx I-2.

frontal is similar to those of other halszkaraptorines[4,17] in that it is vaulted to accommodate a large orbit and has little contribution to the supratemporal fossa. A sharp nuchal crest is formed by the parietal and the squamosal (Supplementary Fig. 2a–e). The latter also produces a shelf that extends over the quadrate head as in other dromaeosaurids[18]. The paroccipital process curves gently on the occiput and has a broad dorsal surface that tapers laterally (Fig. 2f and Supplementary Fig. 2b, e). Its ventrolateral orientation is reminiscent of *Mahakala*[17] but is different from the more horizontal paroccipital process of *Halszkaraptor*[4]. The occipital condyle is long and constricted at its base. A shallow dorsal tympanic recess on the lateral wall of the braincase is different from the deep one of *Mahakala*[17]. The palatine is tetraradiate with a greatly elongated maxillary process, which extends anteriorly beyond the level of the mid-antorbital fenestra. The pterygoid is missing its anterior portion (Fig. 2g and Supplementary Fig. 2a–e). A deep fossa on the medial surface of the thin quadrate

ramus is not seen in any other dromaeosaurids. The mandibles of *Natovenator* preserve most of the elements, especially those on the left side (Fig. 1a, b, d and Supplementary Figs. 1a, 2). Each jaw is characterized by a slender dentary with nearly parallel dorsal and ventral margins, a surangular partially fused with the articular, a distinctive surangular shelf, and a fan-shaped retroarticular process that protrudes dorsomedially. The upper dentition of *Natovenator* is heterodont as the premaxillary teeth are morphologically distinct from the maxillary teeth (Fig. 2a, b, e and Supplementary Fig. 1a, c). There are unusually numerous premaxillary teeth tightly packed without any separation of the alveoli by bony septa. The roots of the teeth are long, and the crowns are tall and incisiviform as in *Halszkaraptor*[4]. Moreover, the large replacement teeth in the premaxilla suggest that the replacement of the premaxillary teeth was delayed as in *Halszkaraptor*[4]. However, the number of teeth in each premaxilla is 13 in *Natovenator*, whereas it is only 11 in *Halszkaraptor*[4].

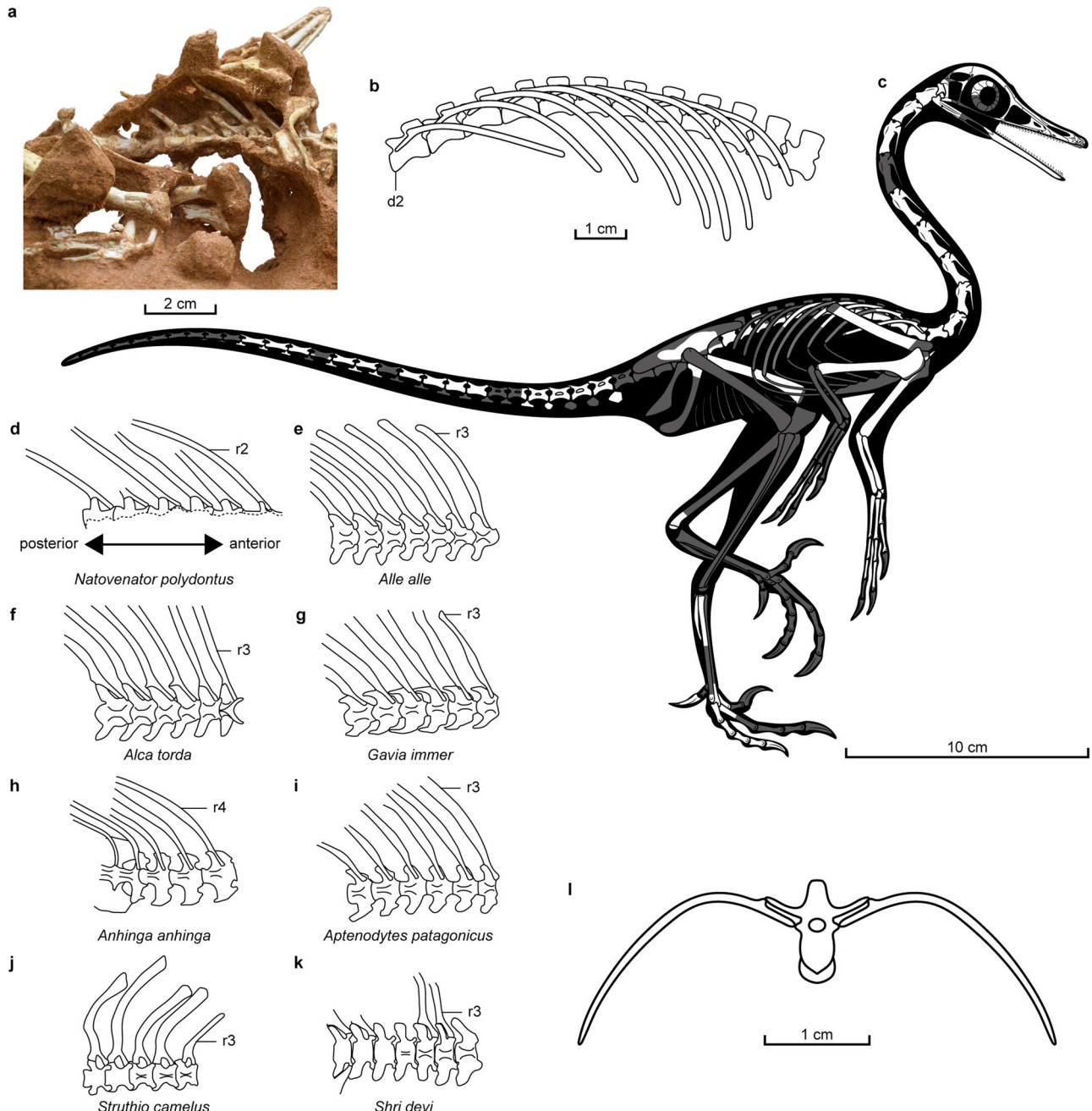

**Fig. 4 Body plan of *Natovenator polydontus* (MPC-D 102/114, holotype) and dorsal rib morphology of various diving birds and terrestrial taxa. a** Dorsal series of *Natovenator* in ventral view. **b** Reconstruction of dorsal vertebrae and ribs of *Natovenator* in left lateral view. **c** Skeletal reconstruction of *Natovenator* with missing parts in dark grey. **d**–**i** Dorsal rib morphology of *Natovenator* (**d**), diving birds (**e**–**i**), common ostrich (**j**), and *Shri devi*, a likely terrestrial dromaeosaurid from the Baruungoyot Formation (**k**) in ventral view (not to scale). **l** Reconstruction of the fourth dorsal vertebra with corresponding ribs in anterior view. d2 second dorsal vertebra, r2 second dorsal rib, r3 third dorsal rib, r4 fourth dorsal rib.

In the maxilla, the three most anterior maxillary teeth are markedly shorter than the premaxillary teeth and the more posterior maxillary teeth. This pattern is also observed in *Halszkaraptor*, although the number of shorter maxillary teeth differs as it has two reduced ones[7]. Both the maxillary and dentary teeth have sharp fang-like crowns that lack serrations. Although posteriormost parts are poorly preserved, there are at least 23 alveoli in each of the maxilla and dentary, which suggests high numbers of teeth in both elements.

The neck of *Natovenator*, as preserved, is twisted and includes ten elongated cervical vertebrae, although most of the 5th cervical is missing (Figs. 1, 3a–d). This elongation of the cervicals results

in a noticeably longer neck than those of most dromaeosaurids and is estimated to be longer than the dorsal series. It is, however, proportionately shorter than that of *Halszkaraptor*, which has a neck as long as its dorsal and sacral vertebra combined[4]. Another peculiarity in the neck of the *Natovenator* is a pronounced posterolaterally extending projection on the neurapophysis of the atlas (Fig. 3a and Supplementary Fig. 2b, c, e). The postzygapophyses of each anterior cervical are fused into a single lobe-like process as in *Halszkaraptor*[4]. Pleurocoels are absent in the cervical vertebrae. In contrast, *Halszkaraptor* has pleurocoels on its 7th–9th cervicals[4]. A total of 12 dorsal vertebrae are preserved (Figs. 1a, b, 3e, 4a and Supplementary Figs. 3a–d). They all lack

pleurocoels, and their parapophyses on the anterior and mid-dorsals are placed high on the anterodorsal end of each centrum. Interestingly, the positions of the parapophyses are similar to those of hesperornithiforms[19–21] rather than other dromaeosaurids such as *Deinonychus*[22] or *Velociraptor*[23]. The preserved dorsal ribs, articulated with the second to seventh dorsals, are flattened and posteriorly oriented (Figs. 1, 3e, 4a–d). The proximal shafts are also nearly horizontal, which is indicative of a dorsoventrally compressed ribcage. Each proximal caudal vertebra has a long centrum and horizontal zygapophyses with expanded laminae (Fig. 3f and Supplementary Fig. 3e–i), all of which are characters shared with other halszkaraptorines[4,17]. The forelimb elements are partially exposed (Figs. 1a, b, 2a–d, 3e, g). The nearly complete right humerus is proportionately short and distally flattened like that of *Halszkaraptor*[4]. The shaft of the ulna is mediolaterally compressed to produce a sharp posterior margin as in *Halszkaraptor*[4] and *Mahakala*[17]. Metacarpal III is robust and is only slightly longer than metacarpal II. Similarly, metacarpal III is almost as thick and long as other second metacarpals of other halszkaraptorines[4,17]. The femur has a long ridge on its posterior surface, which is another characteristic shared among halszkaraptorines[4]. Typically for a dromaeosaurid, metatarsals II and III have ginglymoid distal articular surfaces (Fig. 3h and Supplementary Fig. 4f, h). The ventral surface of metatarsal III is invaded by a ridge near the distal end, unlike other halszkaraptorines (Fig. 3h)[4,5,17,24].

*Phylogenetic analysis.* The phylogenetic analysis found more than 99,999 most parsimonious trees (CI = 0.23, RI = 0.55) with 6574 steps. Deinonychosaurian monophyly is not supported by the strict consensus tree (Supplementary Fig. 6). Instead, Dromaeosauridae was recovered as a sister clade to a monophyletic clade formed by Troodontidae and Avialae, which is consistent with the results of Cau et al.[4] and Cau[7]. Halszkaraptorinae is positioned at the base of Dromaeosauridae as in Cau et al.[4], although there are claims that dromaeosaurid affinities of halszkaraptorines are not well supported[25]. Nine (seven ambiguous and two unambiguous) synapomorphies support the inclusion of Halszkaraptorinae in Dromaeosauridae. The two unambiguous synapomorphies are the anterior tympanic recess at the same level as the basipterygoid process and the presence of a ventral flange on the paroccipital process. A total of 20 synapomorphies (including one unambiguous synapomorphy) unite the four halszkaraptorines, including *Natovenator* (Supplementary Fig. 7). In Halszkaraptorinae, *Halszkaraptor* is the earliest branching taxon, and the remaining three taxa form an unresolved clade supported by three ambiguous synapomorphies (characters 121/1, 569/0, and 1153/1). Two of these synapomorphies are related to

the paroccipital process (characters 121 and 569), which is not preserved in *Hulsanpes*[5,24]. The other is the presence of an expansion on the medial margin of the distal half of metatarsal III, which is not entirely preserved in the *Natovenator*. When scored as 0 for this character, *Natovenator* branches off from the unresolved clade. It suggests that the medial expansion of the dorsal surface of metatarsal III could be a derived character among halszkaraptorines.

**Discussion**

Many anatomical characteristics of *Natovenator* are interpreted here as valuable indicators of this taxon's lifestyle. Specifically, a low and mediolaterally expanded premaxilla with enlarged teeth compared to the posterior dentition, a complex network of neurovascular foramina that is extensively developed on the premaxilla, many teeth in both upper and lower dentitions, a delayed replacement pattern of premaxillary teeth, reduced anterior maxillary teeth, retracted and dorsolaterally facing external nares, a greatly elongated neck, and the horizontal zygapophyses in the cervical and proximal caudal vertebrae are among the ecological indicators shared with *Halszkaraptor* and many reptiles with aquatic adaptations such as plesiosaurians, turtles, and spinosaurids[4,26]. Among them, having retracted nares has been debated that this might not be a proper aquatic adaptation[27]. Another debatable feature is a delayed replacement pattern of premaxillary teeth, which is not directly related to aquatic habits. Although this is shared with sauropterygians, this pattern allows them to keep providing enlarged teeth[28–30]. Based on the relatively large premaxillary teeth of *Natovenator* and *Halszkaraptor*[4], the delayed replacement pattern likely served a similar role. There is also a trend among modern birds that aquatic taxa possess long necks, presumably related to feeding habits and bracing impacts during dives[31]. In the case of the *Natovenator*, the elongated neck might have aided in catching prey rather than in reducing impact because it is unlikely to be able to fly. Additionally, *Natovenator* provides additional insight into its semiaquatic ecology with its dorsal rib morphology. The dorsal ribs of the *Natovenator* are directed posterolaterally to a substantial extent (Figs. 3e, 4a–d). Therefore, the angle between each rib shaft and its associated articulating vertebra is very low, like many diving birds, but in contrast to terrestrial theropods (Fig. 4e–k and Table 1). In these diving birds, backward-oriented ribs aid swimming by making the body more streamlined[32,33]. This is natural because the posterior orientation of the ribs lowers the dorsoventral height of the body and lengthens the ribcage. The resulting long ribcage then contributes to streamlining the body in diving birds[34]. In addition to diving birds, the semiaquatic modern platypus[35] and possible semiaquatic archosauromorph *Tanystropheus*[36] also possess ribs that extend posteriorly. On the other hand, the ribs in fully

**Table 1 Dorsal rib angles of *Natovenator polydontus*, various diving birds, and terrestrial taxa measured in ventral view.**

| | Rib angle (°) | | | | | | | |
|---|---|---|---|---|---|---|---|---|
| **Taxon** | **r2** | **r3** | **r4** | **r5** | **r6** | **r7** | **r8** | **r9** |
| *Natovenator polydontus* | 48 | 40 | 39 | 34 | N/A | 38 | N/A | N/A |
| *Alle alle* | 66 | 70 | 66 | 55 | 47 | 42 | 48 | 43 |
| *Alca torda* | 66 | 67 | 62 | 55 | 51 | 47 | 42 | - |
| *Gavia immer* | 52 | 64 | 58 | 56 | 54 | 52 | 48 | 48 |
| *Anhinga anhinga* | 55 | 60 | 56 | 62 | 70 | 77* | - | - |
| *Aptenodytes patagonicus* | 62 | 64 | 62 | 68 | 66 | 53 | 48 | - |
| *Struthio camelus* | 131 | 134 | 131 | 130 | 114 | 116 | 117 | - |
| *Shri devi* | N/A | 80 | 84 | N/A | N/A | N/A | 89 | N/A |

Proximal rib shaft angles are measured against the vertebral column. Diving birds include *Alle alle*, *Alca torda*, *Gavia immer*, *Anhinga anhinga*, and *Aptenodytes patagonicus*. Terrestrial taxa include *Struthio camelus* and *Shri devi*.
r2–r9 indicate second–ninth dorsal (thoracic) ribs.
*This value is the angle of the first sacral rib.

**Table 2 Rib morphology in *Natovenator polydontus* and other tetrapods with streamlined bodies.**

| Taxon | Rib orientation | Anterior migration of the ribcage | Lifestyle |
|---|---|---|---|
| *Natovenator polydontus* | Posterior | Absent | Semiaquatic |
| Diving birds | Posterior | Absent | Semiaquatic |
| *Ornithorhynchus anatinus* | Posterior | Absent | Semiaquatic |
| Mosasaurs | Slightly posterior | Present | Fully aquatic |
| Extant cetaceans | Slightly posterior | Present | Fully aquatic |

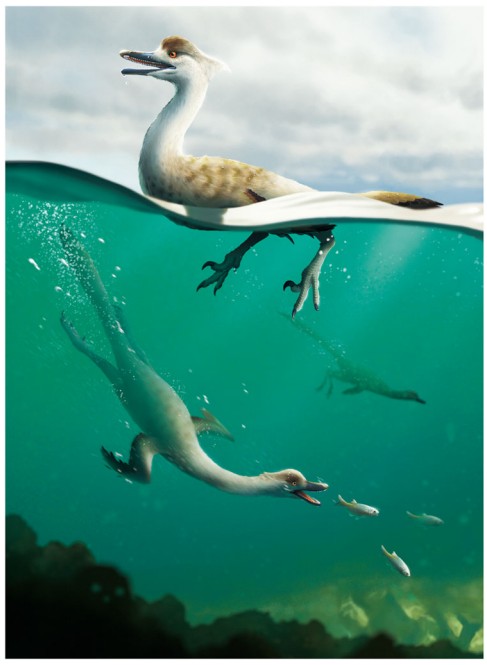

**Fig. 5 Life reconstruction of *Natovenator polydontus* (Artwork by Yusik Choi).** The reconstruction shows the proposed swimming behaviour of *Natovenator polydontus*.

aquatic tetrapods such as mosasaurs and extant cetaceans are posteriorly oriented relative to the long axis of the body parts because of inclined thoracic vertebrae, and the anterior migration of the ribcage and abdominal organs is also instrumental in streamlining their bodies[37–41]. Consequently, *Natovenator* acquired a similar rib profile to that of semiaquatic amniotes (Table 2). Another notable feature of the dorsal ribs of the *Natovenator* is that the proximal shaft forms a wide arch (Fig. 4l), which suggests it had a dorsoventrally compressed ribcage. This barrel-shaped ribcage is also known in putative semiaquatic vertebrates, including spinosaurids[42,43] and choristoderes[44,45]. The rib morphology of *Natovenator* thus implies convergences with various (semi-)aquatic sauropsids and further supports its semiaquatic lifestyle. Also, the streamlined body inferred from the rib configuration strongly indicates that *Natovenator* was a potentially efficient swimmer (Fig. 5). Although the mode of locomotion in water for the *Natovenator* is unknown, based on its close phylogenetic relationship with *Halszkaraptor* (Fig. 3i), forelimbs probably were the primary source of propulsion when swimming, as has been suggested for the latter[4]. Furthermore, the rib morphology of the *Natovenator* helps resolve the debate on the ecology of *Halszkaraptor*[5,46]. Based on the numerous similarities between *Natovenator* and *Halszkaraptor*, it is reasonable to assume that the latter also had a streamlined body and a similar lifestyle. The previous argument that *Halszkaraptor* represents a transitional taxon rather than a semiaquatic one[46] thus can be refuted.

The morphology of the *Natovenator* also provides vital information for understanding the body plan of halszkaraptorines because it has many anatomical characteristics previously restricted to *Halszkaraptor*, including the shared ecological indicators described here. Specifically, the horizontal zygapophyses of the proximal caudal vertebrae are shared with *Mahakala*[17]. It is also notable that *Natovenator* is from the Baruungoyot Formation, whereas *Halszkaraptor* is from the Djadochta beds. The striking similarities between *Natovenator* and *Halszkaraptor* demonstrate that halszkaraptorines in both Baruungoyot and Djadochta formations probably occupied nearly identical ecological niches. The halszkaraptorine body plan may thus be applied to *Hulsanpes*, which is only known from a fragmentary skeleton[5,24]. The streamlined body of the *Natovenator* also reflects the high diversity of body shapes among non-avian dinosaurs and exemplifies convergent evolution with diving birds.

Ever since land vertebrates emerged, many different groups have secondarily adapted to aquatic environments[47]. Dinosaurs have been peculiar in this regard because only avian dinosaurs are known for various aquatic forms, including extinct clades[21]. The body plan of the *Natovenator* makes it clear that some non-avian dinosaurs returned to the water.

## Methods

***µCT scans***. Parts of MPC-D 102/114 were scanned by µCT (or X-ray microscope) to effectively visualize their morphology and internal structures. The skull (excluding its most posterior region) and preserved partial sacrum were scanned via a Skyscan 1276 from Bruker at the Common Research Facility of the School of Biological Sciences at Seoul National University. The back part of the skull with the three anterior cervical vertebrae was scanned by an Xradia 620 Versa from Zeiss at the National Center for Interuniversity Research Facilities at Seoul National University. The parameters used can be found in the Supplementary Information (Supplementary Tables 2, 3). Dragonfly from Object Research Systems was also used in processing the resulting images.

**Phylogenetic analysis**. To investigate the relationships of *Natovenator* with other theropods, a phylogenetic analysis was conducted using a revised data matrix from Cau[7], based on Cau et al.[4]. The modifications that were made in the data matrix are the addition of *Natovenator* (Supplementary Data 1), removal of four taxa (*Alnashetri*, *Shanag*, *Fukuivenator*, and *Hesperornithoides*) to prevent collapses of major clades, two character scorings of *Mahakala* regarding parapophyses of dorsal vertebrae (character 238; from 0 to 1) and the existence of a fibular notch on the calcaneum (character 1430; from ? to 1) based on the description of this taxa from Turner et al.[17]. As a result, 182 taxa with 1807 characters (four ordered) were incorporated in our matrix, then analyzed via TNT ver 1.5[48]. The maximum number of trees was set to 99,999, and *Herrerasaurus* was used as the outgroup taxon. A "New Technology Search" including "Sect. Search" (with RSS, CSS, and XSS checked), "Ratchet," "Drift," and "Tree fusing" was performed with default parameters, followed by the final round of "Traditional Search," also with default parameters, to further explore the shortest trees. Bremer support values at each node were calculated using the Bremer.run script.

**Nomenclatural acts**. This published work and the nomenclatural acts it contains have been registered in ZooBank, the proposed online registration system for the International Code of Zoological Nomenclature (ICZN). The ZooBank LSIDs (Life Science Identifiers) can be resolved and the associated information viewed through any standard web browser by appending the LSID to the prefix "http://zoobank.org/". The LSIDs for this publication are: E50586D4-1135-49B8-9912-3B3A4261CEBF for the genus; 9A6C7438-1B6D-4026-AF55-76B604055EA8 for the species.

**Reporting summary**. Further information on research design is available in the Nature Research Reporting Summary linked to this article.

## Data availability

The character list and scorings (excluding those of *Natovetator*) are available in Cau[7]. The holotype specimen of *Natovenator* (MPC-D 102/114) is housed in the Institute of Paleontology in Ulaanbaatar, Mongolia. The µCT scanned images are deposited at morphosource (https://www.morphosource.org/concern/media/000471331 for the skull excluding the occipital region and https://www.morphosource.org/concern/media/000471343 for the occipital region with anterior cervicals).

## Code availability

The data matrix of the *Natovenator* is included in Supplementary Data 1.

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

## Acknowledgements

Thanks go to all field crew members of the Korea-Mongolia International Dinosaur Expedition (KID) 2008. The KID expedition was supported by a grant to Y.-N.L. from Hwaseong City, Gyeonggi Province, South Korea. We appreciate H.-J. Lee for some of the photographs used in this study and preparation of the specimen with D.K. Kim, W. Kim for help with initial rounds of µCT scanning with Skyscan 1276, M. Choi and M. Lee for technical support with Xradia 620 Versa and Dragonfly software, and the Willi Hennig Society for distribution of TNT version 1.5. M. Son is also greatly appreciated for

his insightful comments. We also thank editor L. R. Grinham and three reviewers, D. Hone, F. L. Agnolin, and T. Holtz Jr. for their constructive comments. Ben Creisler provided helpful opinions on the genus name of the new taxon. This research was supported by Basic Science Research Program through the National Research Foundation of Korea (NRF), funded by the Ministry of Education (grant number 2022R1I1A2060919) to Y.-N.L. and J.-Y.P. (grant number 2022R1A6A3A01085883).

## Author contributions

S.L. processed the μCT data, conducted the phylogenetic analysis, and wrote the manuscript. Y.-N.L. designed and supervised the project. P.J.C. and R.S. discovered and extracted the specimen. R.B. and K.T. provided resources for fieldwork in the Gobi. S.L., J.-Y.P., and S.-H.K. produced the figures. All authors contributed to the discussion and editing of the manuscript.

## Competing interests

The authors declare no competing interests.
