## [Peer Review File · Communications Biology]

Reviewers' comments:

Reviewer #1 (Remarks to the Author):

Overall this manuscript is in very good shape and I have only very minor concerns and comments about it (and not too many of them). See attached marked up document for these.

There is however one major and serious issue that needs to be fully addressed. A major part of the argument about this paper is that the laterally directed ribs is linked to a) streamlining and b) specifically based on the form seen in diving birds. However, support for this contention is essentially non-existent.

On line 204-205 you cite 3 papers to support this contention (you 7, 36 & 37). I checked all three of these and found the following - the first of these only cites the other two as a source for this point about rib angles and so really isn't independent. Both the other two only make simple statements 'rib angle is linked to streamlining and diving in birds' without themselves providing any data or evidence for the point at all. They are unsupported statements without even giving the nature of this observation (e.g. 'our examination of numerous diving birds shows that...').

As such, there is no actual evidence for this at all and if you intend to argue that this animal was a diver based on rib angles this needs actual support and these papers do not provide it. You'll need to actually survey and measure diving birds and non-divers and see if the rib angle does correlate with diving. Even then, I'm not sure you can argue that this is specifically about streamlining without any actual evidence given then a dorsally compressed body should produce more drag than one with a circular cross-section so that needs to be discussed and explored too.

To be clear, I am not saying this is wrong, but it is unsupported and given how central it is to the arguments here this needs to be addressed properly.

Reviewer #2 (Remarks to the Author):

Dear Editor Communications Biology,

I have reviewed the MS entitled "A swimming non-avian dinosaur with a streamlined body" authors Lee et al.

I congratulate the authors for such concise and well-written MS, The new information, as well as the newly described taxon, are very valuable and should be published after minor corrections.

I am attaching a PDF including minor observations on the MS.

All the best,

Federico Agnolin

Reviewer #3 (Remarks to the Author):

The authors describe a new specimen of halzkaraptorine dromaeosaurid, which helps clarify some aspects of the anatomy of this recently discovered and somewhat enigmatic taxon. The new specimen includes skeletal elements not available or not well preserved in previously described individuals.

In particular, the new anatomical features revealed support the earlier hypothesis that halzkaraptorines were adapted to a semi-aquatic mode of life, perhaps more so than other non-avian dinosaurs. Some features of the skull found in Natavenator and in Halskaraptor, for instance, but

lacking in typical dromaeosaurids are also found in water-dwelling birds. It is worth noting, however, that retraction of the external naris by itself is not necessarily an aquatic adaptation, or at least not a subaquatic one. As discussed in Hone and Holtz (2021) it is dorsal placement of the naris rather than posterior retraction that is indicative of a subaqueous mode of locomotion.

Hone, D.W.E., and T.R. Holtz, Jr. 2021. Evaluating the ecology of Spinosaurus: shoreline generalist or aquatic pursuit specialist? *Palaeontologia Electronica* 24(1): a03. doi: 10.26879/1110

The evidence of the morphology and orientation of the dorsal ribs is particularly interesting, since as the authors note this indicates a streamlined body form as in semiaquatic birds.

The detailed anatomical descriptions provided mostly in the Supplementary data, and the phylogenetic analysis, are sufficient to demonstrate this is a new taxon. The illustrations and data tables are all necessary to support the conclusions of the text. That said, a photograph or drawing of the orientation of the ribs in a non-streamlined theropod (especially one from the Baruungoyot) might be useful to contrast the position of those compared to *Natovenator* and the modern aquatic birds.

It might be worthwhile for the authors to describe the geological/sedimentological setting of the discovery and the support for the subaqueous nature of the deposition at the locality. (Given that this is a morphological/phylogenetic paper it is understandable that these geological issues are not the primary focus. However, perhaps in a future work, that information might be provided to help clarify the environment in which this purported aquatic hunter lived.)

A non-scientific, but etymological, question: is the name properly formed? Other compound words I can think of derived from "nato" take the form "nata-" (as in "natatorium" for "swimming pool" or "natatorial" for "adapted to swimming"). You might want to check with etymology and taxonomy expert Ben Creisler (bcreisler@gmail.com) to check, in case you don't have any Latin experts among the team.

A swimming non-avian dinosaur with a streamlined body

[revised manuscript text omitted]

metacarpal II, and the distal part of the shaft of metatarsal III has a slightly pinched ventral
surface.

**Description**

The skull of *Natovenator* is nearly complete, although the preorbital region has been
affected by compression and is slightly offset from the rest of the skull (Figs. 1c, d, 2a–d,
Extended Data Figs. 1, 2). Near the tip of the snout, the premaxilla is marked by a broad
groove. The body of the premaxilla is also dorsoventrally low and is perforated by numerous
foramina that lead into a complex network of neurovascular chambers (Extended Data Fig. 1b)
as in *Halszkaraptor*[9]. Similarly, the external naris is positioned posteriorly and is level with
the premaxilla-maxilla contact (Fig. 2a, b), although it is marginally behind this position in
*Halszkaraptor*[9]. A few characteristics in the snout of *Natovenator* – such as the
exceptionally long external naris and accordingly elongated internarial process (Fig. 2c)
(neither of which are known in *Halszkaraptor* because of poor preservation) – are unique

[revised manuscript text omitted]

indicators of the lifestyle of this taxon. Specifically, a low and mediolaterally expanded
premaxilla with enlarged teeth, a complex network of neurovascular foramina that is
extensively developed on the premaxilla, many teeth in both upper and lower dentitions, a
delayed replacement pattern of premaxillary teeth, reduced anterior maxillary teeth, retracted
external nares, a greatly elongated neck, and the horizontal zygapophyses in the cervical and
proximal caudal vertebrae are among the ecological indicators shared with *Halszkaraptor* and
many reptiles with aquatic adaptations such as plesiosaurians, turtles, and spinosaurids[9, 32].
There is also a trend among modern birds that aquatic taxa possess long necks, presumably
related to feeding habits and bracing impacts during dives[33]. Additionally, *Natovenator*
provides additional insight into its semiaquatic ecology with its dorsal rib morphology. The
dorsal ribs of *Natovenator* are directed posterolaterally to a significant extent (Figs. 3e, 4a–d).
Therefore, the angle between each rib shaft and its associated articulating vertebra is very low
(Extended Data Table 1). A similar condition is known in the semi- or fully aquatic
archosauromorph *Tanystropheus*[34]. Furthermore, certain extant diving birds – such as
alcids[11] and phalacrocoracids [35] – have posteriorly extending ribs as well (Fig. 4e–i). In
these animals, backward-oriented ribs aid swimming by making the body more streamlined[7,
36, 37]. Another notable feature of the dorsal ribs of *Natovenator* is that the proximal shaft
forms a wide arch (Fig. 4j), which suggests it had a dorsoventrally compressed ribcage. This
barrel-shaped ribcage is also known in putative semiaquatic vertebrates, including
spinosaurids[38, 39] and choristoderes[13, 40]. The rib morphology of *Natovenator* thus
implies convergences with various (semi-)aquatic sauropsids and further supports the theory
of its semiaquatic lifestyle. Also, the streamlined body inferred from the rib configuration
strongly indicates that *Natovenator* was an efficient swimmer. Although the mode of
locomotion in water for *Natovenator* is unknown, based on its close phylogenetic relationship
with *Halszkaraptor* (Fig. 3i), forelimbs probably were the primary source of propulsion when
swimming, as has been suggested for the latter[9]. Furthermore, the aquatic adaptations in
*Natovenator* help resolve the debate on the ecology of *Halszkaraptor*[17, 41]. The previous
argument that *Halszkaraptor* represents a transitional taxon rather than a semiaquatic one[41]
can be refuted because the streamlined body and other specializations of *Natovenator*

concretely support the semiaquatic ecology of *Halszkaraptor*.

The morphology of *Natovenator* also provides vital information for understanding
the body plan of halszkaraptorines because it has many anatomical characters previously
restricted to *Halszkaraptor*, including the shared ecological indicators described here. Some
of these traits, especially those in the proximal caudal vertebrae, are shared with *Mahakala*
[24]. It is also notable that *Natovenator* is from the Baruungoyot Formation, whereas
*Halszkaraptor* is from the Djadochta beds. The striking similarities between *Natovenator* and
*Halszkaraptor* demonstrate that halszkaraptorines in both Baruungoyot and Djadochta
formations probably occupied nearly identical ecological niches. The halszkaraptorine body
plan may thus be applied to *Hulsanpes*, which is only known from a fragmentary skeleton [17,
31]. The streamlined body of *Natovenator* also reflects the high diversity of body shapes
among non-avian dinosaurs and exemplifies convergent evolution with diving birds.

Ever since land vertebrates emerged, many different groups have secondarily adapted
to aquatic environments [42]. Dinosaurs have been peculiar in this regard because only avian
dinosaurs are known for various aquatic forms, including extinct clades [28]. The body plan
of *Natovenator* makes it clear that some non-avian dinosaurs returned to the water.

**Methods**

**μ CT Scans**

Parts of MPC-D 102/114 were scanned by μ CT (or X-ray microscope) to effectively visualize
their morphology and internal structures. The skull (excluding its most posterior region) and
preserved partial sacrum were scanned via a Skyscan 1276 from Bruker at the Common
Research Facility of School of Biological Sciences at Seoul National University. The back
part of the skull with the three anterior cervical vertebrae was scanned by an Xradia 620
Versa from Zeiss at the National Center for Interuniversity Research Facilities at Seoul
National University. The parameters used can be found in the Supplementary Information
(Supplementary Tables 2, 3). Dragonfly from Object Research Systems was also used in
processing the resulting images.

**Phylogenetic Analysis**

To investigate the relationships of *Natovenator* with other theropods, a phylogenetic analysis
was conducted using a revised data matrix from Cau[19], which is based on that of Cau et
al.[9]. The modifications that were made in the data matrix are the addition of *Natovenator*,
removal of four taxa (*Alnashetri*, *Shanag*, *Fukuivenator*, and *Hesperornithoides*) to prevent
collapses of major clades, two-character scorings of *Mahakala* regarding parapophyses of
dorsal vertebrae (character 238; from 0 to 1) and the existence of a fibular notch on the
calcaneum (character 1430; from ? to 1) based on the description of this taxa from Turner et
al.[24]. As a result, 182 taxa with 1807 characters (four ordered) were incorporated in our
matrix, then analyzed via TNT ver 1.5[43]. The maximum number of trees was set to 99,999,
and *Herrerasaurus* was used as the outgroup taxon. A "New Technology Search" including
"Sect. Search" (with RSS, CSS, and XSS checked), "Ratchet," "Drift," and "Tree fusing" was
performed with default parameters, followed by the final round of "Traditional Search," also
with default parameters, to further explore the shortest trees. Bremer support values at each
node were calculated using the Bremer.run script.

**References**

- 1. Williams, T. M. Locomotion in the North American mink, a semi-aquatic mammal. I.
Swimming energetics and body drag. *Journal of Experimental Biology* **103**, 155–168
(1983).
- 2. Williams, T. M. & Kooyman, G. L. Swimming performance and hydrodynamic
characteristics of harbor seals *Phoca vitulina*. *Physiological Zoology* **58**, 576–589 (1985).
- 3. Fish, F. E. Influence of hydrodynamic-design and propulsive mode on mammalian
swimming energetics. *Australian Journal of Zoology* **42**, 79–101 (1994).
- 4. Vogel, S. *Life in Moving Fluids: The Physical Biology of Flow* 2nd (Princeton University
Press, 1994).
- 5. Enstipp, M. R., Grémillet, D. & Lorentsen, S.-H. Energetic costs of diving and thermal
status in European shags(*Phalacrocorax aristotelis*). *Journal of Experimental Biology*
**208**, 3451–3461 (2005).
- 6. Tickle, P. G., Lean, S. C., Rose, K. A. R., Wadugodapitiya, A. P. & Codd, J. R. The
influence of load carrying on the energetics and kinematics of terrestrial locomotion in a
diving bird. *Biology Open* **2**, 1239–1244 (2013).

- 7. Brocklehurst, R. J., Schachner, E. R., Codd, J. R. & Sellers, W. I. Respiratory evolution in
archosaurs. *Philosophical Transactions of the Royal Society B: Biological Sciences* **375**,
20190140 (2020).
- 8. Ibrahim, N. *et al.* Semiaquatic adaptations in a giant predatory dinosaur. *Science* **345**,
1613–1616 (2014).
- 9. Cau, A. *et al.* Synchrotron scanning reveals amphibious ecomorphology in a new clade of
bird-like dinosaurs. *Nature* **552**, 395–399 (2017).
- 10. Dyke, G. *et al.* Aerodynamic performance of the feathered dinosaur *Microraptor* and the
evolution of feathered flight. *Nature Communications* **4**, 1–9 (2013).
- 11. Kuroda, N. Morpho-anatomical analysis of parallel evolution between Diving Petrel and
Ancient Auk. *Journal of the Yamashina Institute for Ornithology* **5**, 111–137 (1967).
- 12. Erickson, B. R. Aspects of some anatomical structures of *Champsosaurus* (Reptilia:
Eosuchia). *Journal of Vertebrate Paleontology* **5**, 111–127 (1985).
- 13. Matsumoto, R., Suzuki, S., Tsogtbaatar, K. & Evans, S. E. New material of the enigmatic
reptile *Khurendukhosaurus* (Diapsida: Choristodera) from Mongolia.
*Naturwissenschaften* **96**, 233–242 (2009).
- 14. Norell, M. A., Clark, J. M., Chiappe, L. M. & Dashzeveg, D. A nesting dinosaur. *Nature*
**378**, 774–776 (1995).
- 15. Barsbold, R. *et al.* A pygostyle from a non-avian theropod. *Nature* **403**, 155–156 (2000).
- 16. Lee, Y.-N. *et al.* Resolving the long-standing enigmas of a giant ornithomimosaur
*Deinocheirus mirificus*. *Nature* **515**, 257–260 (2014).
- 17. Osmólska, H. *Hulsanpes perlei* n.g. n.sp. (Deinonychosauria, Saurischia, Dinosauria)
from the Upper Cretaceous Barun Goyot Formation of Mongolia. *Neues Jahrbuch für*
*Geologie und Palaeontologie. Monatshefte*, 440–448 (1982).
- 18. Turner, A. H., Pol, D., Clarke, J. A., Erickson, G. M. & Norell, M. A. A Basal
Dromaeosaurid and Size Evolution Preceding Avian Flight. *Science* **317**, 1378–1381
(2007).
- 19. Cau, A. The body plan of *Halszkaraptor escuilliei* (Dinosauria, Theropoda) is not a
transitional form along the evolution of dromaeosaurid hypercarnivory. *PeerJ* **8**, e8672
(2020).
- 20. Eberth, D. A. Stratigraphy and paleoenvironmental evolution of the dinosaur-rich

- Baruungoyot-Nemegt succession (Upper Cretaceous), Nemegt Basin, southern Mongolia.
*Palaeogeography, Palaeoclimatology, Palaeoecology* **494**, 29–50 (2018).
- 21. Field, D. J. *et al.* Complete *Ichthyornis* skull illuminates mosaic assembly of the avian
head. *Nature* **557**, 96–100 (2018).
- 22. Clark, J. M., Altangerel, P. & Norell, M. A. The skull of *Erlicosaurus andrewsi*, a Late
Cretaceous “Segnosaur” (Theropoda: Therizinosauridae) from Mongolia. *American*
*Museum Novitates*, 1–39 (1994).
- 23. Pu, H. *et al.* An unusual basal therizinosaur dinosaur with an ornithischian dental
arrangement from Northeastern China. *PLOS ONE* **8**, e63423 (2013).
- 24. Turner, A. H., Pol, D. & Norell, M. A. Anatomy of *Mahakala omnogovae* (Theropoda:
Dromaeosauridae), Tögrögiin Shiree, Mongolia. *American Museum Novitates*, 1–66
(2011).
- 25. Norell, M. A. & Makovicky, P. J. in *The Dinosauria* (eds Weishampel, D. B., Dodson, P.
& Osmólska, H.) 2nd, 196–209 (University of California Press, 2004).
- 26. Marsh, O. C. *Odontornithes: a monograph on the extinct toothed birds of North America*.
201 (Washington D.C.: Government Printing Office, 1880).
- 27. Tokaryk, T. T. & Harington, C. R. *Baptornis* sp. (Aves: Hesperornithiformes) from the
Judith River Formation (Campanian) of Saskatchewan, Canada. *Journal of Paleontology*
**66**, 1010–1012 (1992).
- 28. Rees, J. & Lindgren, J. Aquatic birds from the Upper Cretaceous (Lower Campanian) of
Sweden and the biology and distribution of hesperornithiforms. *Palaeontology* **48**, 1321–
1329 (2005).
- 29. Ostrom, J. H. Osteology of *Deinonychus antirrhopus*, an unusual theropod from the
Lower Cretaceous of Montana. *Bulletin of the Peabody Museum of Natural History* **30**,
1–165 (1969).
- 30. Norell, M. A. & Makovicky, P. J. Important features of the dromaeosaurid skeleton II:
Information from newly collected specimens of *Velociraptor mongoliensis*. *American*
*Museum Novitates*, 1–45 (1999).
- 31. Cau, A. & Madzia, D. Redescription and affinities of *Hulsanpes perlei* (Dinosauria,
Theropoda) from the Upper Cretaceous of Mongolia. *PeerJ* **6**, e4868 (2018).
- 32. Fabbri, M. *et al.* Subaqueous foraging among carnivorous dinosaurs. *Nature* **603**, 852–

- 857 (2022).
- 33. Böhmer, C., Plateau, O., Cornette, R. & Abourachid, A. Correlated evolution of neck
length and leg length in birds. *Royal Society Open Science* **6**, 181588 (2019).
- 34. Rieppel, O. *et al.* *Tanystropheus* cf. *T. longobardicus* from the early Late Triassic of
Guizhou Province, southwestern China. *Journal of Vertebrate Paleontology* **30**, 1082–
1089 (2010).
- 35. Shufeldt, R. W. Comparative osteology of Harris's Flightless Cormorant (*Nannopterum*
*harrisi*). *Emu - Austral Ornithology* **15**, 86–114 (1915).
- 36. Tickle, P. G., Ennos, A. R., Lennox, L. E., Perry, S. F. & Codd, J. R. Functional
significance of the uncinata processes in birds. *Journal of Experimental Biology* **210**,
3955–3961 (2007).
- 37. Tickle, P., Nudds, R. & Codd, J. Uncinate process length in birds scales with resting
metabolic rate. *PLOS ONE* **4**, 1–6 (2009).
- 38. Stromer, E. Ergebnisse der Forschungsreisen Prof. E. Stromers in den Wüsten Ägyptens.
II. Wirbeltier-Reste der Baharije-Stufe (unterstes Cenoman). 3. Das Original des
Theropoden *Spinosaurus aegyptiacus* nov. gen., nov. spec. *Abhandlungen der Königlich*
*Bayerischen Akademie der Wissenschaften Mathematisch - physikalische Klasse* **28**, 1–
32 (1915).
- 39. Allain, R., Xaisanavong, T., Richir, P. & Khentavong, B. The first definitive Asian
spinosaurid (Dinosauria: Theropoda) from the Early Cretaceous of Laos.
*Naturwissenschaften* **99**, 369–377 (2012).
- 40. Erickson, B. R. *The lepidosaurian reptile Champsosaurus in North America* 91 (Science
Museum of Minnesota, 1972).
- 41. Brownstein, C. D. *Halszkaraptor escuilliei* and the evolution of the paravian *bauplan*.
*Scientific Reports* **9**, 16455 (2019).
- 42. Uhen, M.D. Evolution of marine mammals: Back to the sea after 300 million years. *The*
*Anatomical Record* **290**, 514–522 (2007).
- 43. Goloboff, P. A. & Catalano, S. A. TNT version 1.5, including a full implementation of
phylogenetic morphometrics. *Cladistics* **32**, 221–238 (2016).

**Acknowledgements:** Thanks go to all field crew members of the Korea-Mongolia

International Dinosaur Expedition (KID) 2008. The KID expedition was supported by a
grant to Y.-N.L. from Hwaseong City, Gyeonggi Province, South Korea. We appreciate
H.-J. Lee for some of the photographs used in this study and preparation of the specimen
with D.K. Kim, W. Kim for help with initial rounds of μ CT scanning with Skyscan 1276,
377 M. Choi and M. Lee for technical support with Xradia 620 Versa and Dragonfly software,
and the Willi Hennig Society for distribution of TNT version 1.5. M. Son is also greatly
appreciated for his insightful comments.

**Author contributions:** S.L processed the μ CT data, conducted the phylogenetic analysis,
and wrote the manuscript. Y.-N.L. designed and supervised the project. P.J.C and R.S.
discovered and extracted the specimen. R.B. and K.T. provided resources for fieldwork
in the Gobi. S.L., J.-Y.P., and S.-H.K. produced the figures. All authors contributed to
the discussion and editing of the manuscript.

**Figure Captions**

**Figure 1. *Natovenator polydontus* (MPC-D 102/114, holotype).** Photographs (**a, c**) and line
drawings (**b, d**) of the main block containing most of the specimen in opposite views.
Abbreviations: cav, caudal vertebra; co, coracoid; cv, cervical vertebra; d, dentary; dc,
distal carpal; dv, dorsal vertebra; fem, femur; fu, furcula; h, humerus; mx, maxilla; ph,
phalanx; pm, premaxilla; r, radius; ul, ulna.

**Figure 2. Skull of *Natovenator polydontus* (MPC-D 102/114, holotype).** **a–d**, Skull in left
lateral (**a**), right lateral (**b**), dorsal (**c**), and ventral (**d**) views. **e**, μ CT-rendered image
sliced at the point marked on **a**, showing a cross-section of the premaxillary and anterior
maxillary teeth in dorsal view. **f**, Micro-computed tomography (μ CT) rendered image of
the occipital region in posterior view. **g**, μ CT rendered image of the pterygoid and
quadrate. Abbreviations: ?bm, possible bite mark; d, dentary; f, frontal; h, humerus; l,
lacrimal; m5, 5th maxillary tooth; mx, maxilla; na, nasal; p, parietal; p13, 13th
premaxillary tooth; pl, palatine; pm, premaxilla; pop, paroccipital process; pt, pterygoid;
q, quadrate; rt, replacement tooth; sq, squamosal; so, supraoccipital.

**Figure 3. Postcranial elements and phylogenetic position of *Natovenator polydontus***
**(MPC-D 102/114, holotype).** **a**, Anterior cervical vertebrae in left lateral view. **b**, Axis
and 3rd cervical vertebra in dorsal view. **c**, 4th cervical vertebra in dorsal view. **d**,
Posterior cervical vertebrae in right lateral view. **e**, Dorsal series in right lateral view. **f**,
Anterior caudal vertebrae in right lateral view. **g**, Left forearm elements in medial view
and manus in ventral view. **h**, Right foot in ventral view. **i**, Phylogenetic position of
*Natovenator* in Dromaeosauridae. Numbers at each node indicate Bremer support values.
Abbreviations: at, atlas; c3, 3rd cervical vertebra; c4, 4th cervical vertebra; c7, 7th
cervical vertebra; c9, 9th cervical vertebra; ch, chevron; d7, 7th dorsal vertebra; fem,
femur; mc I, metacarpal I; mt III, metatarsal III; mt IV, metatarsal IV; poz,
postzygapophysis; prz, prezygapophysis; r, radius; r7, 7th dorsal rib; ul, ulna; I-2, pedal
phalanx I-2.

**Figure 4. Body plan of *Natovenator polydontus* (MPC-D 102/114, holotype) and dorsal**
**rib morphology of various diving birds.** **a**, Dorsal series of *Natovenator* in ventral
view. **b**, Reconstruction of dorsal vertebrae and ribs of *Natovenator* in left lateral view. **c**,
Skeletal reconstruction of *Natovenator* with missing parts in dark grey. **d–i**, Dorsal rib
morphology of *Natovenator* (**d**) and diving birds (**e–i**) in ventral view (not to scale). **j**,
Reconstruction of the 4th dorsal vertebra with corresponding ribs in anterior view.
Abbreviations: d2, 2nd dorsal vertebra; r2, 2nd dorsal rib; r3, 3rd dorsal rib; r4, 4th
dorsal rib.

**Extended Data Figure 1. μ CT rendered images of the skull of *Natovenator polydontus***
**(MPC-D 102/114, holotype).** **a**, Skull in left lateral view. **b**, Slice of the snout region
marked on **a** showing the neurovascular chambers in anterior view. **c**, Slice of the upper
dentition marked on **a** showing the cross-section of the premaxillary and anterior
maxillary teeth in dorsal view. Abbreviations: nvc, neurovascular chamber; m1, 1st
maxillary tooth.

**Extended Data Figure 2. μ CT rendered images of the posterior skull of *Natovenator***

*polydontus* (MPC-D 102/114, holotype). Posterior skull in anterior (a), posterior (b),
left lateral (c), right lateral (d), dorsal (e), and ventral (f) views. Abbreviations: an,
angular; at, atlas; ax, axis; dr, dorsal rib; dtr, dorsal tympanic recess; p, parietal; pop,
paroccipital process; pt, pterygoid; q, quadrate; qj, quadratojugal; oc, occipital condyle;
rp, retroarticular process; sa, surangular; sc, scapula; sq, squamosal; so, supraoccipital.

**Extended Data Figure 3. Sacrum and caudal vertebrae of *Natovenator polydontus***

(MPC-D 102/114, holotype). a–d, Sacrum with mid-caudal vertebrae on top in left
lateral (a), right lateral (b), dorsal (c), and ventral (d) views. e, Anterior caudal vertebrae
in left lateral (e), dorsal (f, g), and ventral (h, i) views. f and h show the anteriormost
vertebrae in the preserved caudal series, whereas g and i are focused on the following
ones. Abbreviations: cav, caudal vertebra; fh, femoral head; i, ilium; ldv, last dorsal
vertebra; poz, postzygapophysis; prz, prezygapophysis; pub, pubis.

**Extended Data Figure 4. Hind limb and pedal elements of *Natovenator polydontus***

(MPC-D 102/114, holotype). a–e, Distal part of the left tibiotarsus in anterior (a),
posterior (b), medial (c), lateral (d), and distal (e) views. f–h, Right foot in medial (f),
lateral (g), and dorsal (h) views. Abbreviations: ap, ascending process of astragalus; mt
III, metatarsal III; mt IV, metatarsal IV; mt V, metatarsal V.

**Extended Data Figure 5. Locality of halszkaraptorines. a, A map of Mongolia. b, A**

magnified map (red rectangle in a) showing occurrences of halszkaraptorines, including
*Natovenator polydontus*. c, The site from which *Natovenator polydontus* (MPC-D
102/114) was recovered.

**Extended Data Figure 6. Strict consensus of the most parsimonious trees found in the**
**phylogenetic analysis.** Numbers at each node indicate Bremer support values.

**Extended Data Figure 7. Phylogeny of Halszkaraptorinae on the strict consensus tree**

**with synapomorphies.** Black circles indicate unambiguous synapomorphies, while
white circles indicate ambiguous ones.

**Extended Data Table 1. Dorsal rib angles in *Natovenator polydontus* and various diving**

**birds measured in ventral view.** Proximal rib shaft angles are measured against the

vertebral column. Abbreviations r2–r9 indicate 2nd–9th dorsal (thoracic) ribs. *This

value is the angle of the first sacral rib.

10 cm

1 To Reviewer #1

Referee comment	Reply
Overall this manuscript is in very good shape and I have only very minor concerns and comments about it (and not too many of them). See attached marked up document for these. There is however one major and serious issue that needs to be fully addressed. A major part of the argument about this paper is that the laterally directed ribs is linked to a) streamlining and b) specifically based on the form seen in diving birds. However, support for this contention is essentially non-existent. On line 204-205 you cite 3 papers to support this contention (you 7, 36 & 37). I checked all three of these and found the following - the first of these only cites the other two as a source for this point about rib angles and so really isn't independent. Both the other two only make simple statements 'rib angle is linked to streamlining and diving in birds' without themselves providing any data or evidence for the point at all. They are unsupported statements without even giving the nature of this observation (e.g. 'our examination of numerous diving birds shows that...). As such, there is no actual evidence for this at all and if you intend to argue that this animal was a diver based on rib angles this needs actual support and these papers do not provide it. You'll need to actually survey and measure diving birds and non-divers and see if the rib angle does correlate with diving. Even then, I'm not sure you can argue that this is specifically about streamlining without any actual	We would like to thank Reviewer #1 for his/her critical comments on our manuscript. Reviewer #1 raised an issue on our interpretation of the dorsal rib orientation related to a streamlined body. Our response to this concern is as follows. First, Reviewer #1 commented “A major part of the argument about this paper is that the laterally directed ribs is linked to a) streamlining and b) specifically based on the form seen in diving birds.” While the ribs are (postero)laterally flared out and make the body dorsoventrally compressed, this is not what we argued that streamlines the body. Instead, the streamlined body is formed by the posterior orientation of the ribs as stated in our manuscript (lines 204–212; The dorsal ribs of Natovenator are directed posterolaterally to a significant extent (Figs. 3e, 4a–d). Therefore, the angle between each rib shaft and its associated articulating vertebra is very low, like many diving birds, but in contrast to terrestrial theropods (Fig. 4e–k, Table 1). A similar condition is known in the semi- or fully aquatic archosauromorph Tanystropheus[32]. Furthermore, certain extant diving birds – such as alcids[33] and phalacrocoracids[34] – also have posteriorly extending ribs. In these animals, backward-oriented ribs aid swimming by making the body more streamlined[35, 36]. It is natural because the posterior orientation of the ribs lowers the dorsoventral height of the body, especially posterior to the middle.). The rib angles show degrees of posterior orientation, not lateral orientation. It would not be streamlining if the ribs were only laterally oriented. Second, Reviewer #1 commented “On line 204-205 you cite 3 papers to support this contention (you 7, 36 & 37). I checked all three of these and found the following - the first of these only cites the other two as a source for this point about rib angles and so really isn't independent. The other two only make simple statements 'rib angle is linked to streamlining and diving in birds' without themselves providing any data or evidence for the point. They are unsupported statements without even giving the nature of this observation (e.g. 'our examination of numerous diving birds shows that...).” We agree that the reference 7 is not independent and removed it. The rib angle in the other two references shows a degree of posterior orientation, as in our examples, and Fig. 1 in Tickle et al. (2007) (reference 35 in the revised manuscript) clearly shows the diving bird razorbill (C) with a more streamlined body has lower rib angles. Posterior orientation of ribs streamlines the body because its dorsoventral height is naturally lowered. We added this as the following sentence (lines 211–212; This is natural because the posterior orientation of the ribs lowers the dorsoventral height of the body, especially posterior to the middle.).

evidence given then a dorsally compressed body should produce more drag than one with a circular cross-section so that needs to be discussed and explored too. To be clear, I am not saying this is wrong, but it is unsupported and given how central it is to the arguments here this needs to be addressed properly.	Third, Reviewer #1 commented “You’ll need to actually survey and measure diving birds and non-divers and see if the rib angle does correlate with diving.” We actually did measure several diving birds, and the rib angles are included in Table 1 (originally Extended Data Table 1). These birds have a streamlined body with posteriorly oriented ribs (lower rib angle). We added rib morphology and angles of some non-divers (Struthio camelus, the common ostrich and Shri devi, a dromaeosaurid from the Baruungoyot Formation) that do not have streamlined bodies to Fig. 4 (j, k) and Table 1, respectively. Lastly, Reviewer #1 commented “Even then, I’m not sure you can argue that this is specifically about streamlining without any actual evidence given then a dorsally compressed body should produce more drag than one with a circular cross-section, so that needs to be discussed and explored too.” Again, we did not state the streamlining results just from a dorsoventral compression. Additionally, Reviewer #1’s statement “a dorsally compressed body should produce more drag than one with a circular cross-section” is not true because a circle is always broader than an ellipse given the same circumference, hence less drag on the latter. The dorsoventral compression of the body, therefore, reduces drag, and various aquatic vertebrates have a dorsoventrally compressed body, as stated in our manuscript (lines 214–216; This barrel-shaped ribcage is also known in putative semiaquatic vertebrates, including spinosaurids[37, 38] and choristoderes[39, 40].).
Comments in the attached pdf	Responses to these comments are included in the attachment pdf (1_reviewer_attachment_1_1655289959_convrt_response.pdf).

2

3 **To Reviewer #2**

Referee comment	Reply
I have reviewed the MS entitled "A swimming non-avian dinosaur with a streamlined body" authors Lee et al. I congratulate the authors for such concise and well-written MS, The new information, as well as the newly described taxon, are very valuable and should be published after minor corrections. I am attaching a PDF including minor observations on the MS.	We appreciate the careful review and comments from Reviewer #2. Responses to the comments are included in the attachment pdf (2_reviewer_attachment_1_1656378962_convrt_response.pdf).

4

5 **To Reviewer #3**

Referee comment	Reply
The authors describe a new specimen of	We thank Reviewer#3 for his/her valuable

halszkaraptorine dromaeosaurid, which helps clarify some aspects of the anatomy of this recently discovered and somewhat enigmatic taxon. The new specimen includes skeletal elements not available or not well preserved in previously described individuals. In particular, the new anatomical features revealed support the earlier hypothesis that halszkaraptorines were adapted to a semi-aquatic mode of life, perhaps more so than other non-avian dinosaurs. Some features of the skull found in Natavenator and in Halszkaraptor, for instance, but lacking in typical dromaeosaurids are also found in water-dwelling birds. It is worth noting, however, that retraction of the external naris by itself is not necessarily an aquatic adaptation, or at least not a subaquatic one. As discussed in Hone and Holtz (2021) it is dorsal placement of the naris rather than posterior retraction that is indicative of a subaqueous mode of locomotion. Hone, D.W.E., and T.R. Holtz, Jr. 2021. Evaluating the ecology of Spinosaurus: shoreline generalist of aquatic pursuit specialist? Palaeontologia Electronica 24(1): a03. doi: 10.26879/1110	comments on our manuscript. Following the Reviewer#3's comment regarding the nares, we added their relative dorsal placement to the description and discussion sections of the main manuscript (lines 58 (dorsolateral placement of retracted external nares), 98 and 99 (It is also dorsally placed compared to those of other non-avian theropods and faces dorsolaterally.), and 191 (retracted and dorsolaterally facing external nares)).
The evidence of the morphology and orientation of the dorsal ribs is particularly interesting, since as the authors note this indicates a streamlined body form as in semiaquatic birds. The detailed anatomical descriptions provided mostly in the Supplementary data, and the phylogenetic analysis, are sufficient to demonstrate this is a new taxon. The illustrations and data tables are all necessary to support the conclusions of the text. That said, a photograph or drawing of the orientation of the ribs in a non-streamlined theropod (especially one from the Baruungoyot) might be useful to contrast the position of those compared to Natavenator and the modern aquatic birds.	Following this comment of Reviewer #3, we added a likely terrestrial (not streamlined) dromaeosaurid theropod (Shri devi from the Baruungoyot Formation) to Fig. 4 (Fig. 4k).
It might be worthwhile for the authors to describe the geological/sedimentological setting of the discovery and the support for the subaqueous nature of the deposition at the locality. (Given that this is a morphological/phylogenetic paper it is understandable that these geological issues are not the primary focus. However, perhaps in a future work, that information might be provided to help clarify the environment in which this purported aquatic hunter lived.)	We cited Eberth (2018) (reference 13), which gives information on the sedimentology and palaeoenvironment of the Baruungoyot Formation (as well as the Nemegt Fm.). As of now, there is no particularly new sedimentological information we can add to our manuscript. As Reviewer #3 commented, however, it will be worthwhile to examine the site that yielded the new specimen and gather more information on the geological setting.

A non-scientific, but etymological, question: is the name properly formed? Other compound words I can think of derived from "nato" take the form "nata-" (as in "natatorium" for "swimming pool" or "natatorial" for "adapted to swimming). You might want to check with etymology and taxonomy expert Ben Creisler (bcreisler@gmail.com) to check, in case you don't have any Latin experts among the team.	We consulted Ben Creisler as suggested, but he said that Natovenator should be fine. To quote him, he said "The spelling Natovenator to mean "swimming hunter" from the Latin verb nato "swim" should be fine. It is similar to some other generic names formed with a Latin verb. A spelling "natavenator" would not work in this case.". Following his reply, we decided to use the name 'Natovenator' as it is.
---	---

A swimming non-avian dinosaur with a streamlined body

[revised manuscript text omitted]

**Diagnosis.** A small halszkaraptorine dromaeosaurid with the following **unique combination**
**of characters (autapomorphies among dromaeosaurids with asterisks):** wide groove delimited

by a pair of ridges on the anterodorsal surface of the premaxilla*, premaxilla with an
elongated internarial process that overlies nasal and extends posterior to the external naris*,
13 premaxillary teeth* whose crowns are incisiviform and mostly larger than the maxillary or
dentary teeth, first three anteriormost maxillary teeth are significantly reduced and are
clustered together with the following tooth without any separations by interdental septa*,
anteroposteriorly long external naris (about 30% of the preorbital skull length)*, paroccipital
process with a anteroposteriorly broad dorsal surface*, elongate maxillary process of the
palatine that extends anteriorly beyond the middle of the antorbital fenestra*, pterygoid with
a deep fossa on the medial surface of the quadrate ramus*, distinct posterolaterally oriented
projection on the lateral surface of atlas*, absence of pleurocoels in cervical vertebrae (not
confirmed in the missing 5th cervical centrum)*, posterolaterally oriented and nearly
horizontal proximal shafts in the dorsal ribs*, hourglass-shaped metacarpal II with distinctly
concave medial and lateral surfaces*, metacarpal III is longer but more slender than

metacarpal II, and the distal part of the shaft of metatarsal III has a slightly pinched ventral
surface.

**Description**

The skull of *Natovenator* is nearly complete, although the preorbital region has been
affected by compression and is slightly offset from the rest of the skull (Figs. 1c, d, 2a–d,
Extended Data Figs. 1, 2). Near the tip of the snout, the premaxilla is marked by a broad
groove. The body of the premaxilla is also dorsoventrally low and is perforated by numerous
foramina that lead into a complex network of neurovascular chambers (Extended Data Fig. 1b)
as in *Halszkaraptor*[9]. Similarly, the external naris is positioned posteriorly and is level with
the premaxilla-maxilla contact (Fig. 2a, b), although it is marginally behind this position in
*Halszkaraptor*[9]. A few characteristics in the snout of *Natovenator* – such as the
exceptionally long external naris and accordingly elongated internarial process (Fig. 2c)
(neither of which are known in *Halszkaraptor* because of poor preservation) – are unique

[revised manuscript text omitted]

indicators of the lifestyle of this taxon. Specifically, a low and mediolaterally expanded
premaxilla with enlarged teeth, a complex network of neurovascular foramina that is
extensively developed on the premaxilla, many teeth in both upper and lower dentitions, a
delayed replacement pattern of premaxillary teeth, reduced anterior maxillary teeth, retracted
external nares, a greatly elongated neck, and the horizontal zygapophyses in the cervical and
proximal caudal vertebrae are among the ecological indicators shared with *Halszkaraptor* and
many reptiles with aquatic adaptations such as plesiosaurians, turtles, and spinosaurids[9, 32].
There is also a trend among modern birds that aquatic taxa possess long necks, presumably
related to feeding habits and bracing impacts during dives[33]. Additionally, *Natovenator*
provides additional insight into its semiaquatic ecology with its dorsal rib morphology. The
dorsal ribs of *Natovenator* are directed posterolaterally to a significant extent (Figs. 3e, 4a–d).
Therefore, the angle between each rib shaft and its associated articulating vertebra is very low
(Extended Data Table 1). A similar condition is known in the semi- or fully aquatic
archosauromorph *Tanystropheus*[34]. Furthermore, certain extant diving birds – such as
alcids[11] and phalacrocoracids [35] – have posteriorly extending ribs as well (Fig. 4e–i). In
these animals, backward-oriented ribs aid swimming by making the body more streamlined[7,
36, 37]. Another notable feature of the dorsal ribs of *Natovenator* is that the proximal shaft
forms a wide arch (Fig. 4j), which suggests it had a dorsoventrally compressed ribcage. This
barrel-shaped ribcage is also known in putative semiaquatic vertebrates, including
spinosaurids[38, 39] and choristoderes[13, 40]. The rib morphology of *Natovenator* thus
implies convergences with various (semi-)aquatic sauropsids and further supports the theory
of its semiaquatic lifestyle. Also, the streamlined body inferred from the rib configuration
strongly indicates that *Natovenator* was an efficient swimmer. Although the mode of
locomotion in water for *Natovenator* is unknown, based on its close phylogenetic relationship
with *Halszkaraptor* (Fig. 3i), forelimbs probably were the primary source of propulsion when
swimming, as has been suggested for the latter[9]. Furthermore, the aquatic adaptations in
*Natovenator* help resolve the debate on the ecology of *Halszkaraptor*[17, 41]. The previous
argument that *Halszkaraptor* represents a transitional taxon rather than a semiaquatic one[41]
can be refuted because the streamlined body and other specializations of *Natovenator*

concretely support the semiaquatic ecology of *Halszkaraptor*.

The morphology of *Natovenator* also provides vital information for understanding
the body plan of halszkaraptorines because it has many anatomical characters previously
restricted to *Halszkaraptor*, including the shared ecological indicators described here. **Some**
**of these traits, especially those in the proximal caudal vertebrae,** are shared with *Mahakala*
[24]. It is also notable that *Natovenator* is from the Baruungoyot Formation, whereas
*Halszkaraptor* is from the Djadochta beds. The striking similarities between *Natovenator* and
*Halszkaraptor* demonstrate that halszkaraptorines in both Baruungoyot and Djadochta
formations probably occupied nearly identical ecological niches. The halszkaraptorine body
plan may thus be applied to *Hulsanpes*, which is only known from a fragmentary skeleton [17,
31]. The streamlined body of *Natovenator* also reflects the high diversity of body shapes
among non-avian dinosaurs and exemplifies convergent evolution with diving birds.

Ever since land vertebrates emerged, many different groups have secondarily adapted
to aquatic environments [42]. Dinosaurs have been peculiar in this regard because only avian
dinosaurs are known for various aquatic forms, including extinct clades [28]. The body plan
of *Natovenator* makes it clear that some non-avian dinosaurs returned to the water.

**Methods**

**μ CT Scans**

Parts of MPC-D 102/114 were scanned by μ CT (or X-ray microscope) to effectively visualize
their morphology and internal structures. The skull (excluding its most posterior region) and
preserved partial sacrum were scanned via a Skyscan 1276 from Bruker at the Common
Research Facility of School of Biological Sciences at Seoul National University. The back
part of the skull with the three anterior cervical vertebrae was scanned by an Xradia 620
Versa from Zeiss at the National Center for Interuniversity Research Facilities at Seoul
National University. The parameters used can be found in the Supplementary Information
(Supplementary Tables 2, 3). Dragonfly from Object Research Systems was also used in
processing the resulting images.

**Phylogenetic Analysis**

To investigate the relationships of *Natovenator* with other theropods, a phylogenetic analysis
was conducted using a revised data matrix from Cau[19], which is based on that of Cau et
al.[9]. The modifications that were made in the data matrix are the addition of *Natovenator*,
removal of four taxa (*Alnashetri*, *Shanag*, *Fukuivenator*, and *Hesperornithoides*) to prevent
collapses of major clades, two-character scorings of *Mahakala* regarding parapophyses of
dorsal vertebrae (character 238; from 0 to 1) and the existence of a fibular notch on the
calcaneum (character 1430; from ? to 1) based on the description of this taxa from Turner et
al.[24]. As a result, 182 taxa with 1807 characters (four ordered) were incorporated in our
matrix, then analyzed via TNT ver 1.5[43]. The maximum number of trees was set to 99,999,
and *Herrerasaurus* was used as the outgroup taxon. A "New Technology Search" including
"Sect. Search" (with RSS, CSS, and XSS checked), "Ratchet," "Drift," and "Tree fusing" was
performed with default parameters, followed by the final round of "Traditional Search," also
with default parameters, to further explore the shortest trees. Bremer support values at each
node were calculated using the Bremer.run script.

**References**

- 1. Williams, T. M. Locomotion in the North American mink, a semi-aquatic mammal. I.
Swimming energetics and body drag. *Journal of Experimental Biology* **103**, 155–168
(1983).
- 2. Williams, T. M. & Kooyman, G. L. Swimming performance and hydrodynamic
characteristics of harbor seals *Phoca vitulina*. *Physiological Zoology* **58**, 576–589 (1985).
- 3. Fish, F. E. Influence of hydrodynamic-design and propulsive mode on mammalian
swimming energetics. *Australian Journal of Zoology* **42**, 79–101 (1994).
- 4. Vogel, S. *Life in Moving Fluids: The Physical Biology of Flow* 2nd (Princeton University
Press, 1994).
- 5. Enstipp, M. R., Grémillet, D. & Lorentsen, S.-H. Energetic costs of diving and thermal
status in European shags(*Phalacrocorax aristotelis*). *Journal of Experimental Biology*
**208**, 3451–3461 (2005).
- 6. Tickle, P. G., Lean, S. C., Rose, K. A. R., Wadugodapitiya, A. P. & Codd, J. R. The
influence of load carrying on the energetics and kinematics of terrestrial locomotion in a
diving bird. *Biology Open* **2**, 1239–1244 (2013).

- 7. Brocklehurst, R. J., Schachner, E. R., Codd, J. R. & Sellers, W. I. Respiratory evolution in
archosaurs. *Philosophical Transactions of the Royal Society B: Biological Sciences* **375**,
20190140 (2020).
- 8. Ibrahim, N. *et al.* Semiaquatic adaptations in a giant predatory dinosaur. *Science* **345**,
1613–1616 (2014).
- 9. Cau, A. *et al.* Synchrotron scanning reveals amphibious ecomorphology in a new clade of
bird-like dinosaurs. *Nature* **552**, 395–399 (2017).
- 10. Dyke, G. *et al.* Aerodynamic performance of the feathered dinosaur *Microraptor* and the
evolution of feathered flight. *Nature Communications* **4**, 1–9 (2013).
- 11. Kuroda, N. Morpho-anatomical analysis of parallel evolution between Diving Petrel and
Ancient Auk. *Journal of the Yamashina Institute for Ornithology* **5**, 111–137 (1967).
- 12. Erickson, B. R. Aspects of some anatomical structures of *Champsosaurus* (Reptilia:
Eosuchia). *Journal of Vertebrate Paleontology* **5**, 111–127 (1985).
- 13. Matsumoto, R., Suzuki, S., Tsogtbaatar, K. & Evans, S. E. New material of the enigmatic
reptile *Khurendukhosaurus* (Diapsida: Choristodera) from Mongolia.
*Naturwissenschaften* **96**, 233–242 (2009).
- 14. Norell, M. A., Clark, J. M., Chiappe, L. M. & Dashzeveg, D. A nesting dinosaur. *Nature*
**378**, 774–776 (1995).
- 15. Barsbold, R. *et al.* A pygostyle from a non-avian theropod. *Nature* **403**, 155–156 (2000).
- 16. Lee, Y.-N. *et al.* Resolving the long-standing enigmas of a giant ornithomimosaur
*Deinocheirus mirificus*. *Nature* **515**, 257–260 (2014).
- 17. Osmólska, H. *Hulsanpes perlei* n.g. n.sp. (Deinonychosauria, Saurischia, Dinosauria)
from the Upper Cretaceous Barun Goyot Formation of Mongolia. *Neues Jahrbuch für*
*Geologie und Palaeontologie. Monatshefte*, 440–448 (1982).
- 18. Turner, A. H., Pol, D., Clarke, J. A., Erickson, G. M. & Norell, M. A. A Basal
Dromaeosaurid and Size Evolution Preceding Avian Flight. *Science* **317**, 1378–1381
(2007).
- 19. Cau, A. The body plan of *Halszkaraptor escuilliei* (Dinosauria, Theropoda) is not a
transitional form along the evolution of dromaeosaurid hypercarnivory. *PeerJ* **8**, e8672
(2020).
- 20. Eberth, D. A. Stratigraphy and paleoenvironmental evolution of the dinosaur-rich

- Baruungoyot-Nemegt succession (Upper Cretaceous), Nemegt Basin, southern Mongolia.
*Palaeogeography, Palaeoclimatology, Palaeoecology* **494**, 29–50 (2018).
- 21. Field, D. J. *et al.* Complete *Ichthyornis* skull illuminates mosaic assembly of the avian
head. *Nature* **557**, 96–100 (2018).
- 22. Clark, J. M., Altangerel, P. & Norell, M. A. The skull of *Erlicosaurus andrewsi*, a Late
Cretaceous “Segnosaur” (Theropoda: Therizinosauridae) from Mongolia. *American*
*Museum Novitates*, 1–39 (1994).
- 23. Pu, H. *et al.* An unusual basal therizinosaur dinosaur with an ornithischian dental
arrangement from Northeastern China. *PLOS ONE* **8**, e63423 (2013).
- 24. Turner, A. H., Pol, D. & Norell, M. A. Anatomy of *Mahakala omnogovae* (Theropoda:
Dromaeosauridae), Tögrögiin Shiree, Mongolia. *American Museum Novitates*, 1–66
(2011).
- 25. Norell, M. A. & Makovicky, P. J. in *The Dinosauria* (eds Weishampel, D. B., Dodson, P.
& Osmólska, H.) 2nd, 196–209 (University of California Press, 2004).
- 26. Marsh, O. C. *Odontornithes: a monograph on the extinct toothed birds of North America*.
201 (Washington D.C.: Government Printing Office, 1880).
- 27. Tokaryk, T. T. & Harington, C. R. *Baptornis* sp. (Aves: Hesperornithiformes) from the
Judith River Formation (Campanian) of Saskatchewan, Canada. *Journal of Paleontology*
**66**, 1010–1012 (1992).
- 28. Rees, J. & Lindgren, J. Aquatic birds from the Upper Cretaceous (Lower Campanian) of
Sweden and the biology and distribution of hesperornithiforms. *Palaeontology* **48**, 1321–
1329 (2005).
- 29. Ostrom, J. H. Osteology of *Deinonychus antirrhopus*, an unusual theropod from the
Lower Cretaceous of Montana. *Bulletin of the Peabody Museum of Natural History* **30**,
1–165 (1969).
- 30. Norell, M. A. & Makovicky, P. J. Important features of the dromaeosaurid skeleton II:
Information from newly collected specimens of *Velociraptor mongoliensis*. *American*
*Museum Novitates*, 1–45 (1999).
- 31. Cau, A. & Madzia, D. Redescription and affinities of *Hulsanpes perlei* (Dinosauria,
Theropoda) from the Upper Cretaceous of Mongolia. *PeerJ* **6**, e4868 (2018).
- 32. Fabbri, M. *et al.* Subaqueous foraging among carnivorous dinosaurs. *Nature* **603**, 852–

- 857 (2022).
- 33. Böhmer, C., Plateau, O., Cornette, R. & Abourachid, A. Correlated evolution of neck
length and leg length in birds. *Royal Society Open Science* **6**, 181588 (2019).
- 34. Rieppel, O. *et al.* *Tanystropheus* cf. *T. longobardicus* from the early Late Triassic of
Guizhou Province, southwestern China. *Journal of Vertebrate Paleontology* **30**, 1082–
1089 (2010).
- 35. Shufeldt, R. W. Comparative osteology of Harris's Flightless Cormorant (*Nannopterum*
*harrisi*). *Emu - Austral Ornithology* **15**, 86–114 (1915).
- 36. Tickle, P. G., Ennos, A. R., Lennox, L. E., Perry, S. F. & Codd, J. R. Functional
significance of the uncinatate processes in birds. *Journal of Experimental Biology* **210**,
3955–3961 (2007).
- 37. Tickle, P., Nudds, R. & Codd, J. Uncinatus process length in birds scales with resting
metabolic rate. *PLOS ONE* **4**, 1–6 (2009).
- 38. Stromer, E. Ergebnisse der Forschungsreisen Prof. E. Stromers in den Wüsten Ägyptens.
II. Wirbeltier-Reste der Baharije-Stufe (unterstes Cenoman). 3. Das Original des
Theropoden *Spinosaurus aegyptiacus* nov. gen., nov. spec. *Abhandlungen der Königlich*
*Bayerischen Akademie der Wissenschaften Mathematisch - physikalische Klasse* **28**, 1–
32 (1915).
- 39. Allain, R., Xaisanavong, T., Richir, P. & Khentavong, B. The first definitive Asian
spinosaurid (Dinosauria: Theropoda) from the Early Cretaceous of Laos.
*Naturwissenschaften* **99**, 369–377 (2012).
- 40. Erickson, B. R. *The lepidosaurian reptile Champsosaurus in North America* 91 (Science
Museum of Minnesota, 1972).
- 41. Brownstein, C. D. *Halszkaraptor escuilliei* and the evolution of the paravian *bauplan*.
*Scientific Reports* **9**, 16455 (2019).
- 42. Uhen, M.D. Evolution of marine mammals: Back to the sea after 300 million years. *The*
*Anatomical Record* **290**, 514–522 (2007).
- 43. Goloboff, P. A. & Catalano, S. A. TNT version 1.5, including a full implementation of
phylogenetic morphometrics. *Cladistics* **32**, 221–238 (2016).

**Acknowledgements:** Thanks go to all field crew members of the Korea-Mongolia

International Dinosaur Expedition (KID) 2008. The KID expedition was supported by a
grant to Y.-N.L. from Hwaseong City, Gyeonggi Province, South Korea. We appreciate
H.-J. Lee for some of the photographs used in this study and preparation of the specimen
with D.K. Kim, W. Kim for help with initial rounds of μ CT scanning with Skyscan 1276,
377 M. Choi and M. Lee for technical support with Xradia 620 Versa and Dragonfly software,
and the Willi Hennig Society for distribution of TNT version 1.5. M. Son is also greatly
appreciated for his insightful comments.

**Author contributions:** S.L processed the μ CT data, conducted the phylogenetic analysis,
and wrote the manuscript. Y.-N.L. designed and supervised the project. P.J.C and R.S.
discovered and extracted the specimen. R.B. and K.T. provided resources for fieldwork
in the Gobi. S.L., J.-Y.P., and S.-H.K. produced the figures. All authors contributed to
the discussion and editing of the manuscript.

**Figure Captions**

**Figure 1. *Natovenator polydontus* (MPC-D 102/114, holotype).** Photographs (**a, c**) and line
drawings (**b, d**) of the main block containing most of the specimen in opposite views.
Abbreviations: cav, caudal vertebra; co, coracoid; cv, cervical vertebra; d, dentary; dc,
distal carpal; dv, dorsal vertebra; fem, femur; fu, furcula; h, humerus; mx, maxilla; ph,
phalanx; pm, premaxilla; r, radius; ul, ulna.

**Figure 2. Skull of *Natovenator polydontus* (MPC-D 102/114, holotype).** **a–d**, Skull in left
lateral (**a**), right lateral (**b**), dorsal (**c**), and ventral (**d**) views. **e**, μ CT-rendered image
sliced at the point marked on **a**, showing a cross-section of the premaxillary and anterior
maxillary teeth in dorsal view. **f**, Micro-computed tomography (μ CT) rendered image of
the occipital region in posterior view. **g**, μ CT rendered image of the pterygoid and
quadrate. Abbreviations: ?bm, possible bite mark; d, dentary; f, frontal; h, humerus; l,
lacrimal; m5, 5th maxillary tooth; mx, maxilla; na, nasal; p, parietal; p13, 13th
premaxillary tooth; pl, palatine; pm, premaxilla; pop, paroccipital process; pt, pterygoid;
q, quadrate; rt, replacement tooth; sq, squamosal; so, supraoccipital.

**Figure 3. Postcranial elements and phylogenetic position of *Natovenator polydontus***
**(MPC-D 102/114, holotype).** **a**, Anterior cervical vertebrae in left lateral view. **b**, Axis
and 3rd cervical vertebra in dorsal view. **c**, 4th cervical vertebra in dorsal view. **d**,
Posterior cervical vertebrae in right lateral view. **e**, Dorsal series in right lateral view. **f**,
Anterior caudal vertebrae in right lateral view. **g**, Left forearm elements in medial view
and manus in ventral view. **h**, Right foot in ventral view. **i**, Phylogenetic position of
*Natovenator* in Dromaeosauridae. Numbers at each node indicate Bremer support values.
Abbreviations: at, atlas; c3, 3rd cervical vertebra; c4, 4th cervical vertebra; c7, 7th
cervical vertebra; c9, 9th cervical vertebra; ch, chevron; d7, 7th dorsal vertebra; fem,
femur; mc I, metacarpal I; mt III, metatarsal III; mt IV, metatarsal IV; poz,
postzygapophysis; prz, prezygapophysis; r, radius; r7, 7th dorsal rib; ul, ulna; I-2, pedal
phalanx I-2.

**Figure 4. Body plan of *Natovenator polydontus* (MPC-D 102/114, holotype) and dorsal**
**rib morphology of various diving birds.** **a**, Dorsal series of *Natovenator* in ventral
view. **b**, Reconstruction of dorsal vertebrae and ribs of *Natovenator* in left lateral view. **c**,
Skeletal reconstruction of *Natovenator* with missing parts in dark grey. **d–i**, Dorsal rib
morphology of *Natovenator* (**d**) and diving birds (**e–i**) in ventral view (not to scale). **j**,
Reconstruction of the 4th dorsal vertebra with corresponding ribs in anterior view.
Abbreviations: d2, 2nd dorsal vertebra; r2, 2nd dorsal rib; r3, 3rd dorsal rib; r4, 4th
dorsal rib.

**Extended Data Figure 1. μ CT rendered images of the skull of *Natovenator polydontus***
**(MPC-D 102/114, holotype).** **a**, Skull in left lateral view. **b**, Slice of the snout region
marked on **a** showing the neurovascular chambers in anterior view. **c**, Slice of the upper
dentition marked on **a** showing the cross-section of the premaxillary and anterior
maxillary teeth in dorsal view. Abbreviations: nvc, neurovascular chamber; m1, 1st
maxillary tooth.

**Extended Data Figure 2. μ CT rendered images of the posterior skull of *Natovenator***

*polydontus* (MPC-D 102/114, holotype). Posterior skull in anterior (a), posterior (b),
left lateral (c), right lateral (d), dorsal (e), and ventral (f) views. Abbreviations: an,
angular; at, atlas; ax, axis; dr, dorsal rib; dtr, dorsal tympanic recess; p, parietal; pop,
paroccipital process; pt, pterygoid; q, quadrate; qj, quadratojugal; oc, occipital condyle;
rp, retroarticular process; sa, surangular; sc, scapula; sq, squamosal; so, supraoccipital.

**Extended Data Figure 3. Sacrum and caudal vertebrae of *Natovenator polydontus***

(MPC-D 102/114, holotype). a–d, Sacrum with mid-caudal vertebrae on top in left
lateral (a), right lateral (b), dorsal (c), and ventral (d) views. e, Anterior caudal vertebrae
in left lateral (e), dorsal (f, g), and ventral (h, i) views. f and h show the anteriormost
vertebrae in the preserved caudal series, whereas g and i are focused on the following
ones. Abbreviations: cav, caudal vertebra; fh, femoral head; i, ilium; ldv, last dorsal
vertebra; poz, postzygapophysis; prz, prezygapophysis; pub, pubis.

**Extended Data Figure 4. Hind limb and pedal elements of *Natovenator polydontus***

(MPC-D 102/114, holotype). a–e, Distal part of the left tibiotarsus in anterior (a),
posterior (b), medial (c), lateral (d), and distal (e) views. f–h, Right foot in medial (f),
lateral (g), and dorsal (h) views. Abbreviations: ap, ascending process of astragalus; mt
III, metatarsal III; mt IV, metatarsal IV; mt V, metatarsal V.

**Extended Data Figure 5. Locality of halszkaraptorines. a, A map of Mongolia. b, A**

magnified map (red rectangle in a) showing occurrences of halszkaraptorines, including
*Natovenator polydontus*. c, The site from which *Natovenator polydontus* (MPC-D
102/114) was recovered.

**Extended Data Figure 6. Strict consensus of the most parsimonious trees found in the**
**phylogenetic analysis.** Numbers at each node indicate Bremer support values.

**Extended Data Figure 7. Phylogeny of Halszkaraptorinae on the strict consensus tree**

**with synapomorphies.** Black circles indicate unambiguous synapomorphies, while
white circles indicate ambiguous ones.

**Extended Data Table 1. Dorsal rib angles in *Natovenator polydontus* and various diving**

**birds measured in ventral view.** Proximal rib shaft angles are measured against the

vertebral column. Abbreviations r2–r9 indicate 2nd–9th dorsal (thoracic) ribs. *This

value is the angle of the first sacral rib.

10 cm

A swimming non-avian dinosaur with a streamlined body

[revised manuscript text omitted]

metacarpal II, and the distal part of the shaft of metatarsal III has a slightly pinched ventral
surface.

**Description**

The skull of *Natovenator* is nearly complete, although the preorbital region has been
affected by compression and is slightly offset from the rest of the skull (Figs. 1c, d, 2a–d,
Extended Data Figs. 1, 2). Near the tip of the snout, the premaxilla is marked by a broad
groove. The body of the premaxilla is also dorsoventrally low and is perforated by numerous
foramina that lead into a complex network of neurovascular chambers (Extended Data Fig. 1b)
as in *Halszkaraptor*[9]. Similarly, the external naris is positioned posteriorly and is level with
the premaxilla-maxilla contact (Fig. 2a, b), although it is marginally behind this position in
*Halszkaraptor*[9]. A few characteristics in the snout of *Natovenator* – such as the
exceptionally long external naris and accordingly elongated internarial process (Fig. 2c)
(neither of which are known in *Halszkaraptor* because of poor preservation) – are unique

[revised manuscript text omitted]

indicators of the lifestyle of this taxon. Specifically, a low and mediolaterally expanded
premaxilla with enlarged teeth, a complex network of neurovascular foramina that is
extensively developed on the premaxilla, many teeth in both upper and lower dentitions, a
delayed replacement pattern of premaxillary teeth, reduced anterior maxillary teeth, retracted
external nares, a greatly elongated neck, and the horizontal zygapophyses in the cervical and
proximal caudal vertebrae are among the ecological indicators shared with *Halszkaraptor* and
many reptiles with aquatic adaptations such as plesiosaurians, turtles, and spinosaurids[9, 32].
There is also a trend among modern birds that aquatic taxa possess long necks, presumably
related to feeding habits and bracing impacts during dives[33]. Additionally, *Natovenator*
provides additional insight into its semiaquatic ecology with its dorsal rib morphology. The
dorsal ribs of *Natovenator* are directed posterolaterally to a significant extent (Figs. 3e, 4a–d).
Therefore, the angle between each rib shaft and its associated articulating vertebra is very low
(Extended Data Table 1). A similar condition is known in the semi- or fully aquatic
archosauromorph *Tanystropheus*[34]. Furthermore, certain extant diving birds – such as
alcids[11] and phalacrocoracids [35] – have posteriorly extending ribs as well (Fig. 4e–i). In
these animals, backward-oriented ribs aid swimming by making the body more streamlined[7,
36, 37]. Another notable feature of the dorsal ribs of *Natovenator* is that the proximal shaft
forms a wide arch (Fig. 4j), which suggests it had a dorsoventrally compressed ribcage. This
barrel-shaped ribcage is also known in putative semiaquatic vertebrates, including
spinosaurids[38, 39] and choristoderes[13, 40]. The rib morphology of *Natovenator* thus
implies convergences with various (semi-)aquatic sauropsids and further supports the theory
of its semiaquatic lifestyle. Also, the streamlined body inferred from the rib configuration
strongly indicates that *Natovenator* was an efficient swimmer. Although the mode of
locomotion in water for *Natovenator* is unknown, based on its close phylogenetic relationship
with *Halszkaraptor* (Fig. 3i), forelimbs probably were the primary source of propulsion when
swimming, as has been suggested for the latter[9]. Furthermore, the aquatic adaptations in
*Natovenator* help resolve the debate on the ecology of *Halszkaraptor*[17, 41]. The previous
argument that *Halszkaraptor* represents a transitional taxon rather than a semiaquatic one[41]
can be refuted because the streamlined body and other specializations of *Natovenator*

concretely support the semiaquatic ecology of *Halszkaraptor*.

The morphology of *Natovenator* also provides vital information for understanding
the body plan of halszkaraptorines because it has many anatomical characters previously
restricted to *Halszkaraptor*, including the shared ecological indicators described here. Some
of these traits, especially those in the proximal caudal vertebrae, are shared with *Mahakala*
[24]. It is also notable that *Natovenator* is from the Baruungoyot Formation, whereas
*Halszkaraptor* is from the Djadochta beds. The striking similarities between *Natovenator* and
*Halszkaraptor* demonstrate that halszkaraptorines in both Baruungoyot and Djadochta
formations probably occupied nearly identical ecological niches. The halszkaraptorine body
plan may thus be applied to *Hulsanpes*, which is only known from a fragmentary skeleton [17,
31]. The streamlined body of *Natovenator* also reflects the high diversity of body shapes
among non-avian dinosaurs and exemplifies convergent evolution with diving birds.

Ever since land vertebrates emerged, many different groups have secondarily adapted
to aquatic environments [42]. Dinosaurs have been peculiar in this regard because only avian
dinosaurs are known for various aquatic forms, including extinct clades [28]. The body plan
of *Natovenator* makes it clear that some non-avian dinosaurs returned to the water.

**Methods**

**μ CT Scans**

Parts of MPC-D 102/114 were scanned by μ CT (or X-ray microscope) to effectively visualize
their morphology and internal structures. The skull (excluding its most posterior region) and
preserved partial sacrum were scanned via a Skyscan 1276 from Bruker at the Common
Research Facility of School of Biological Sciences at Seoul National University. The back
part of the skull with the three anterior cervical vertebrae was scanned by an Xradia 620
Versa from Zeiss at the National Center for Interuniversity Research Facilities at Seoul
National University. The parameters used can be found in the Supplementary Information
(Supplementary Tables 2, 3). Dragonfly from Object Research Systems was also used in
processing the resulting images.

**Phylogenetic Analysis**

To investigate the relationships of *Natovenator* with other theropods, a phylogenetic analysis
was conducted using a revised data matrix from Cau[19], which is based on that of Cau et
al.[9]. The modifications that were made in the data matrix are the addition of *Natovenator*,
removal of four taxa (*Alnashetri*, *Shanag*, *Fukuivenator*, and *Hesperornithoides*) to prevent
collapses of major clades, two-character scorings of *Mahakala* regarding parapophyses of
dorsal vertebrae (character 238; from 0 to 1) and the existence of a fibular notch on the
calcaneum (character 1430; from ? to 1) based on the description of this taxa from Turner et
al.[24]. As a result, 182 taxa with 1807 characters (four ordered) were incorporated in our
matrix, then analyzed via TNT ver 1.5[43]. The maximum number of trees was set to 99,999,
and *Herrerasaurus* was used as the outgroup taxon. A "New Technology Search" including
"Sect. Search" (with RSS, CSS, and XSS checked), "Ratchet," "Drift," and "Tree fusing" was
performed with default parameters, followed by the final round of "Traditional Search," also
with default parameters, to further explore the shortest trees. Bremer support values at each
node were calculated using the Bremer.run script.

**References**

- 1. Williams, T. M. Locomotion in the North American mink, a semi-aquatic mammal. I.
Swimming energetics and body drag. *Journal of Experimental Biology* **103**, 155–168
(1983).
- 2. Williams, T. M. & Kooyman, G. L. Swimming performance and hydrodynamic
characteristics of harbor seals *Phoca vitulina*. *Physiological Zoology* **58**, 576–589 (1985).
- 3. Fish, F. E. Influence of hydrodynamic-design and propulsive mode on mammalian
swimming energetics. *Australian Journal of Zoology* **42**, 79–101 (1994).
- 4. Vogel, S. *Life in Moving Fluids: The Physical Biology of Flow* 2nd (Princeton University
Press, 1994).
- 5. Enstipp, M. R., Grémillet, D. & Lorentsen, S.-H. Energetic costs of diving and thermal
status in European shags(*Phalacrocorax aristotelis*). *Journal of Experimental Biology*
**208**, 3451–3461 (2005).
- 6. Tickle, P. G., Lean, S. C., Rose, K. A. R., Wadugodapitiya, A. P. & Codd, J. R. The
influence of load carrying on the energetics and kinematics of terrestrial locomotion in a
diving bird. *Biology Open* **2**, 1239–1244 (2013).

- 7. Brocklehurst, R. J., Schachner, E. R., Codd, J. R. & Sellers, W. I. Respiratory evolution in
archosaurs. *Philosophical Transactions of the Royal Society B: Biological Sciences* **375**,
20190140 (2020).
- 8. Ibrahim, N. *et al.* Semiaquatic adaptations in a giant predatory dinosaur. *Science* **345**,
1613–1616 (2014).
- 9. Cau, A. *et al.* Synchrotron scanning reveals amphibious ecomorphology in a new clade of
bird-like dinosaurs. *Nature* **552**, 395–399 (2017).
- 10. Dyke, G. *et al.* Aerodynamic performance of the feathered dinosaur *Microraptor* and the
evolution of feathered flight. *Nature Communications* **4**, 1–9 (2013).
- 11. Kuroda, N. Morpho-anatomical analysis of parallel evolution between Diving Petrel and
Ancient Auk. *Journal of the Yamashina Institute for Ornithology* **5**, 111–137 (1967).
- 12. Erickson, B. R. Aspects of some anatomical structures of *Champsosaurus* (Reptilia:
Eosuchia). *Journal of Vertebrate Paleontology* **5**, 111–127 (1985).
- 13. Matsumoto, R., Suzuki, S., Tsogtbaatar, K. & Evans, S. E. New material of the enigmatic
reptile *Khurendukhosaurus* (Diapsida: Choristodera) from Mongolia.
*Naturwissenschaften* **96**, 233–242 (2009).
- 14. Norell, M. A., Clark, J. M., Chiappe, L. M. & Dashzeveg, D. A nesting dinosaur. *Nature*
**378**, 774–776 (1995).
- 15. Barsbold, R. *et al.* A pygostyle from a non-avian theropod. *Nature* **403**, 155–156 (2000).
- 16. Lee, Y.-N. *et al.* Resolving the long-standing enigmas of a giant ornithomimosaur
*Deinocheirus mirificus*. *Nature* **515**, 257–260 (2014).
- 17. Osmólska, H. *Hulsanpes perlei* n.g. n.sp. (Deinonychosauria, Saurischia, Dinosauria)
from the Upper Cretaceous Barun Goyot Formation of Mongolia. *Neues Jahrbuch für*
*Geologie und Palaeontologie. Monatshefte*, 440–448 (1982).
- 18. Turner, A. H., Pol, D., Clarke, J. A., Erickson, G. M. & Norell, M. A. A Basal
Dromaeosaurid and Size Evolution Preceding Avian Flight. *Science* **317**, 1378–1381
(2007).
- 19. Cau, A. The body plan of *Halszkaraptor escuilliei* (Dinosauria, Theropoda) is not a
transitional form along the evolution of dromaeosaurid hypercarnivory. *PeerJ* **8**, e8672
(2020).
- 20. Eberth, D. A. Stratigraphy and paleoenvironmental evolution of the dinosaur-rich

- Baruungoyot-Nemegt succession (Upper Cretaceous), Nemegt Basin, southern Mongolia.
*Palaeogeography, Palaeoclimatology, Palaeoecology* **494**, 29–50 (2018).
- 21. Field, D. J. *et al.* Complete *Ichthyornis* skull illuminates mosaic assembly of the avian
head. *Nature* **557**, 96–100 (2018).
- 22. Clark, J. M., Altangerel, P. & Norell, M. A. The skull of *Erlicosaurus andrewsi*, a Late
Cretaceous “Segnosaur” (Theropoda: Therizinosauridae) from Mongolia. *American*
*Museum Novitates*, 1–39 (1994).
- 23. Pu, H. *et al.* An unusual basal therizinosaur dinosaur with an ornithischian dental
arrangement from Northeastern China. *PLOS ONE* **8**, e63423 (2013).
- 24. Turner, A. H., Pol, D. & Norell, M. A. Anatomy of *Mahakala omnogovae* (Theropoda:
Dromaeosauridae), Tögrögiin Shiree, Mongolia. *American Museum Novitates*, 1–66
(2011).
- 25. Norell, M. A. & Makovicky, P. J. in *The Dinosauria* (eds Weishampel, D. B., Dodson, P.
& Osmólska, H.) 2nd, 196–209 (University of California Press, 2004).
- 26. Marsh, O. C. *Odontornithes: a monograph on the extinct toothed birds of North America*.
201 (Washington D.C.: Government Printing Office, 1880).
- 27. Tokaryk, T. T. & Harington, C. R. *Baptornis* sp. (Aves: Hesperornithiformes) from the
Judith River Formation (Campanian) of Saskatchewan, Canada. *Journal of Paleontology*
**66**, 1010–1012 (1992).
- 28. Rees, J. & Lindgren, J. Aquatic birds from the Upper Cretaceous (Lower Campanian) of
Sweden and the biology and distribution of hesperornithiforms. *Palaeontology* **48**, 1321–
1329 (2005).
- 29. Ostrom, J. H. Osteology of *Deinonychus antirrhopus*, an unusual theropod from the
Lower Cretaceous of Montana. *Bulletin of the Peabody Museum of Natural History* **30**,
1–165 (1969).
- 30. Norell, M. A. & Makovicky, P. J. Important features of the dromaeosaurid skeleton II:
Information from newly collected specimens of *Velociraptor mongoliensis*. *American*
*Museum Novitates*, 1–45 (1999).
- 31. Cau, A. & Madzia, D. Redescription and affinities of *Hulsanpes perlei* (Dinosauria,
Theropoda) from the Upper Cretaceous of Mongolia. *PeerJ* **6**, e4868 (2018).
- 32. Fabbri, M. *et al.* Subaqueous foraging among carnivorous dinosaurs. *Nature* **603**, 852–

- 857 (2022).
- 33. Böhmer, C., Plateau, O., Cornette, R. & Abourachid, A. Correlated evolution of neck
length and leg length in birds. *Royal Society Open Science* **6**, 181588 (2019).
- 34. Rieppel, O. *et al.* *Tanystropheus* cf. *T. longobardicus* from the early Late Triassic of
Guizhou Province, southwestern China. *Journal of Vertebrate Paleontology* **30**, 1082–
1089 (2010).
- 35. Shufeldt, R. W. Comparative osteology of Harris's Flightless Cormorant (*Nannopterum*
*harrisi*). *Emu - Austral Ornithology* **15**, 86–114 (1915).
- 36. Tickle, P. G., Ennos, A. R., Lennox, L. E., Perry, S. F. & Codd, J. R. Functional
significance of the uncinata processes in birds. *Journal of Experimental Biology* **210**,
3955–3961 (2007).
- 37. Tickle, P., Nudds, R. & Codd, J. Uncinate process length in birds scales with resting
metabolic rate. *PLOS ONE* **4**, 1–6 (2009).
- 38. Stromer, E. Ergebnisse der Forschungsreisen Prof. E. Stromers in den Wüsten Ägyptens.
II. Wirbeltier-Reste der Baharije-Stufe (unterstes Cenoman). 3. Das Original des
Theropoden *Spinosaurus aegyptiacus* nov. gen., nov. spec. *Abhandlungen der Königlich*
*Bayerischen Akademie der Wissenschaften Mathematisch - physikalische Klasse* **28**, 1–
32 (1915).
- 39. Allain, R., Xaisanavong, T., Richir, P. & Khentavong, B. The first definitive Asian
spinosaurid (Dinosauria: Theropoda) from the Early Cretaceous of Laos.
*Naturwissenschaften* **99**, 369–377 (2012).
- 40. Erickson, B. R. *The lepidosaurian reptile Champsosaurus in North America* 91 (Science
Museum of Minnesota, 1972).
- 41. Brownstein, C. D. *Halszkaraptor escuilliei* and the evolution of the paravian *bauplan*.
*Scientific Reports* **9**, 16455 (2019).
- 42. Uhen, M.D. Evolution of marine mammals: Back to the sea after 300 million years. *The*
*Anatomical Record* **290**, 514–522 (2007).
- 43. Goloboff, P. A. & Catalano, S. A. TNT version 1.5, including a full implementation of
phylogenetic morphometrics. *Cladistics* **32**, 221–238 (2016).

**Acknowledgements:** Thanks go to all field crew members of the Korea-Mongolia

International Dinosaur Expedition (KID) 2008. The KID expedition was supported by a
grant to Y.-N.L. from Hwaseong City, Gyeonggi Province, South Korea. We appreciate
H.-J. Lee for some of the photographs used in this study and preparation of the specimen
with D.K. Kim, W. Kim for help with initial rounds of μ CT scanning with Skyscan 1276,
377 M. Choi and M. Lee for technical support with Xradia 620 Versa and Dragonfly software,
and the Willi Hennig Society for distribution of TNT version 1.5. M. Son is also greatly
appreciated for his insightful comments.

**Author contributions:** S.L processed the μ CT data, conducted the phylogenetic analysis,
and wrote the manuscript. Y.-N.L. designed and supervised the project. P.J.C and R.S.
discovered and extracted the specimen. R.B. and K.T. provided resources for fieldwork
in the Gobi. S.L., J.-Y.P., and S.-H.K. produced the figures. All authors contributed to
the discussion and editing of the manuscript.

**Figure Captions**

**Figure 1. *Natovenator polydontus* (MPC-D 102/114, holotype).** Photographs (**a, c**) and line
drawings (**b, d**) of the main block containing most of the specimen in opposite views.
Abbreviations: cav, caudal vertebra; co, coracoid; cv, cervical vertebra; d, dentary; dc,
distal carpal; dv, dorsal vertebra; fem, femur; fu, furcula; h, humerus; mx, maxilla; ph,
phalanx; pm, premaxilla; r, radius; ul, ulna.

**Figure 2. Skull of *Natovenator polydontus* (MPC-D 102/114, holotype).** **a–d**, Skull in left
lateral (**a**), right lateral (**b**), dorsal (**c**), and ventral (**d**) views. **e**, μ CT-rendered image
sliced at the point marked on **a**, showing a cross-section of the premaxillary and anterior
maxillary teeth in dorsal view. **f**, Micro-computed tomography (μ CT) rendered image of
the occipital region in posterior view. **g**, μ CT rendered image of the pterygoid and
quadrate. Abbreviations: ?bm, possible bite mark; d, dentary; f, frontal; h, humerus; l,
lacrimal; m5, 5th maxillary tooth; mx, maxilla; na, nasal; p, parietal; p13, 13th
premaxillary tooth; pl, palatine; pm, premaxilla; pop, paroccipital process; pt, pterygoid;
q, quadrate; rt, replacement tooth; sq, squamosal; so, supraoccipital.

**Figure 3. Postcranial elements and phylogenetic position of *Natovenator polydontus***
**(MPC-D 102/114, holotype).** **a**, Anterior cervical vertebrae in left lateral view. **b**, Axis
and 3rd cervical vertebra in dorsal view. **c**, 4th cervical vertebra in dorsal view. **d**,
Posterior cervical vertebrae in right lateral view. **e**, Dorsal series in right lateral view. **f**,
Anterior caudal vertebrae in right lateral view. **g**, Left forearm elements in medial view
and manus in ventral view. **h**, Right foot in ventral view. **i**, Phylogenetic position of
*Natovenator* in Dromaeosauridae. Numbers at each node indicate Bremer support values.
Abbreviations: at, atlas; c3, 3rd cervical vertebra; c4, 4th cervical vertebra; c7, 7th
cervical vertebra; c9, 9th cervical vertebra; ch, chevron; d7, 7th dorsal vertebra; fem,
femur; mc I, metacarpal I; mt III, metatarsal III; mt IV, metatarsal IV; poz,
postzygapophysis; prz, prezygapophysis; r, radius; r7, 7th dorsal rib; ul, ulna; I-2, pedal
phalanx I-2.

**Figure 4. Body plan of *Natovenator polydontus* (MPC-D 102/114, holotype) and dorsal**
**rib morphology of various diving birds.** **a**, Dorsal series of *Natovenator* in ventral
view. **b**, Reconstruction of dorsal vertebrae and ribs of *Natovenator* in left lateral view. **c**,
Skeletal reconstruction of *Natovenator* with missing parts in dark grey. **d–i**, Dorsal rib
morphology of *Natovenator* (**d**) and diving birds (**e–i**) in ventral view (not to scale). **j**,
Reconstruction of the 4th dorsal vertebra with corresponding ribs in anterior view.
Abbreviations: d2, 2nd dorsal vertebra; r2, 2nd dorsal rib; r3, 3rd dorsal rib; r4, 4th
dorsal rib.

**Extended Data Figure 1. μ CT rendered images of the skull of *Natovenator polydontus***
**(MPC-D 102/114, holotype).** **a**, Skull in left lateral view. **b**, Slice of the snout region
marked on **a** showing the neurovascular chambers in anterior view. **c**, Slice of the upper
dentition marked on **a** showing the cross-section of the premaxillary and anterior
maxillary teeth in dorsal view. Abbreviations: nvc, neurovascular chamber; m1, 1st
maxillary tooth.

**Extended Data Figure 2. μ CT rendered images of the posterior skull of *Natovenator***

*polydontus* (MPC-D 102/114, holotype). Posterior skull in anterior (a), posterior (b),
left lateral (c), right lateral (d), dorsal (e), and ventral (f) views. Abbreviations: an,
angular; at, atlas; ax, axis; dr, dorsal rib; dtr, dorsal tympanic recess; p, parietal; pop,
paroccipital process; pt, pterygoid; q, quadrate; qj, quadratojugal; oc, occipital condyle;
rp, retroarticular process; sa, surangular; sc, scapula; sq, squamosal; so, supraoccipital.

**Extended Data Figure 3. Sacrum and caudal vertebrae of *Natovenator polydontus***

(MPC-D 102/114, holotype). a–d, Sacrum with mid-caudal vertebrae on top in left
lateral (a), right lateral (b), dorsal (c), and ventral (d) views. e, Anterior caudal vertebrae
in left lateral (e), dorsal (f, g), and ventral (h, i) views. f and h show the anteriormost
vertebrae in the preserved caudal series, whereas g and i are focused on the following
ones. Abbreviations: cav, caudal vertebra; fh, femoral head; i, ilium; ldv, last dorsal
vertebra; poz, postzygapophysis; prz, prezygapophysis; pub, pubis.

**Extended Data Figure 4. Hind limb and pedal elements of *Natovenator polydontus***

(MPC-D 102/114, holotype). a–e, Distal part of the left tibiotarsus in anterior (a),
posterior (b), medial (c), lateral (d), and distal (e) views. f–h, Right foot in medial (f),
lateral (g), and dorsal (h) views. Abbreviations: ap, ascending process of astragalus; mt
III, metatarsal III; mt IV, metatarsal IV; mt V, metatarsal V.

**Extended Data Figure 5. Locality of halszkaraptorines. a, A map of Mongolia. b, A**

magnified map (red rectangle in a) showing occurrences of halszkaraptorines, including
*Natovenator polydontus*. c, The site from which *Natovenator polydontus* (MPC-D
102/114) was recovered.

**Extended Data Figure 6. Strict consensus of the most parsimonious trees found in the**
**phylogenetic analysis.** Numbers at each node indicate Bremer support values.

**Extended Data Figure 7. Phylogeny of Halszkaraptorinae on the strict consensus tree**

**with synapomorphies.** Black circles indicate unambiguous synapomorphies, while
white circles indicate ambiguous ones.

**Extended Data Table 1. Dorsal rib angles in *Natovenator polydontus* and various diving**

**birds measured in ventral view.** Proximal rib shaft angles are measured against the

vertebral column. Abbreviations r2–r9 indicate 2nd–9th dorsal (thoracic) ribs. *This

value is the angle of the first sacral rib.

10 cm

Reviewers' comments:

Reviewer #1 (Remarks to the Author):

I have only one major issue still and that is the idea of rib angle being linked to diving.

I am happy with the idea that the ribs you have here are orientated like those of diving ducks and auks. What you did not demonstrate before, and still have not demonstrated, is that the angle of these ribs directly contributes to streamlining. The only evidence you were able to provide was the three references all of which essentially stated that this was true but themselves without evidence. There is no actual direct evidence in the literature (e.g. a formal study of the distribution of rib shape and behavior, or that compressed ribs increases streamlining) and therefore this remains an unsupported assertion, not anything with actual evidence. You say for example that the razorbill has the more streamlined body in Fig 1 of Tickle et al. but how is this demonstrated? Again there is no actual evidence, you can't look at a skeleton and declare it more streamlined. Even the correlation here hasn't been reasonably demonstrated. Pointing out that an ostrich and dromaeosaur have different ribs really doesn't add to this argument.

To be clear, I strongly suspect you are correct. But the interpretation of halszkaraptorines as being semi-aquatic has been contentious and this is a major argument being put forwards for this behavior and the evidence you have for this is, to my mind, extraordinarily weak. It's pointing to a handful of very grossly similar animals that have some putative behaviours in common and stating that X directly leads to Y and you simply don't have the evidence to support that. You need a much stronger case and I don't think it's unreasonable to ask you do a serious survey of the literature and / or some actual specimens and demonstrate this association rather than infer it from a couple of papers that themselves state it without support.

This needs fixing.

1 To Reviewer #1

Referee comments	Reply
I have only one major issue still and that is the idea of rib angle being linked to diving. I am happy with the idea that the ribs you have here are orientated like those of diving ducks and auks. What you did not demonstrate before, and still have not demonstrated, is that the angle of these ribs directly contributes to streamlining. The only evidence you were able to provide was the three references all of which essentially stated that this was true but themselves without evidence. There is no actual direct evidence in the literature (e.g. a formal study of the distribution of rib shape and behavior, or that compressed ribs increases streamlining) and therefore this remains an unsupported assertion, not anything with actual evidence. You say for example that the razorbill has the more streamlined body in Fig 1 of Tickle et al. but how is this demonstrated? Again there is no actual evidence, you can't look at a skeleton and declare it more streamlined. Even the correlation here hasn't been reasonably demonstrated. Pointing out that an ostrich and dromaeosaur have different ribs really doesn't add to this argument. To be clear, I strongly suspect you are correct. But the interpretation of halszkaraptorines as being semi-aquatic has been contentious and this is a major argument being put forwards for this behavior and the evidence you have for this is, to my mind, extraordinarily weak. It's pointing to a handful of very grossly similar animals that have some putative behaviours in common and stating that X directly leads to Y and you simply don't have the evidence to support that. You need a much stronger case and I don't think it's unreasonable to ask you do a serious survey of the literature and / or some actual specimens and demonstrate this association rather than infer it from a couple of papers that themselves state it without support. This needs fixing.	Our response to the comments from Reviewer #1 is as follows. In our previous response and revised manuscript, we stated how posteriorly oriented ribs contribute to streamlining the body (by lowering the dorsoventral height). A streamlined body is a posteriorly tapering body, and the posterior orientation of ribs, in contrast to the vertical orientation, naturally helps make this shape. Of course, not all aquatic animals have ribs with strong posterior orientations, but as far as we know, all animals with posteriorly oriented ribs have hydrodynamic body profiles. Reviewer #1 commented, "You say, for example, that the razorbill has the more streamlined body in Fig 1 of Tickle et al., but how is this demonstrated?". Because the bird is extant, and it is known that it has a streamlined body, the skeleton helps us understand how they have such a body shape. We previously provided several clades of diving birds as examples for comparison. Although we believe what we presented was good enough, we added more examples and elaboration (lines 208–217: This is natural because the posterior orientation of the ribs lowers the dorsoventral height of the body and lengthens the rib cage. The resulted long rib cage then contributes to streamlining the body in diving birds[34]. In addition to diving birds, the semiaquatic modern platypus[35] and possible semiaquatic archosauromorph Tanystropheus[36] also possess ribs that extend posteriorly. On the other hand, the ribs in fully aquatic tetrapods such as mosasaurs and extant cetaceans are posteriorly oriented relative to the long axis of the body partly because of inclined thoracic vertebrae, and the anterior migration of the rib cage and abdominal organs is also instrumental in streamlining their bodies[37–41]. Consequently, Natovenator acquired a similar rib profile to that of semiaquatic amniotes (Table 2).).